# Mitotic clustering of pulverized chromosomes from micronuclei

Yu-Fen Lin[1], Qing Hu[1,7], Alice Mazzagatti[1,7], Jose Espejo Valle-Inclán[2,7], Elizabeth G. Maurais[1], Rashmi Dahiya[1], Alison Guyer[1,5], Jacob T. Sanders[1,6], Justin L. Engel[1], Giaochau Nguyen[1], Daniel Bronder[3], Samuel F. Bakhoum[3], Isidro Cortés-Ciriano[2 ✉] & Peter Ly[1,4 ✉]

Complex genome rearrangements can be generated by the catastrophic pulverization of missegregated chromosomes trapped within micronuclei through a process known as chromothripsis[1–5]. As each chromosome contains a single centromere, it remains unclear how acentric fragments derived from shattered chromosomes are inherited between daughter cells during mitosis[6]. Here we tracked micronucleated chromosomes with live-cell imaging and show that acentric fragments cluster in close spatial proximity throughout mitosis for asymmetric inheritance by a single daughter cell. Mechanistically, the CIP2A–TOPBP1 complex prematurely associates with DNA lesions within ruptured micronuclei during interphase, which poises pulverized chromosomes for clustering upon mitotic entry. Inactivation of CIP2A–TOPBP1 caused acentric fragments to disperse throughout the mitotic cytoplasm, stochastically partition into the nucleus of both daughter cells and aberrantly misaccumulate as cytoplasmic DNA. Mitotic clustering facilitates the reassembly of acentric fragments into rearranged chromosomes lacking the extensive DNA copy-number losses that are characteristic of canonical chromothripsis. Comprehensive analysis of pan-cancer genomes revealed clusters of DNA copy-number-neutral rearrangements—termed balanced chromothripsis—across diverse tumour types resulting in the acquisition of known cancer driver events. Thus, distinct patterns of chromothripsis can be explained by the spatial clustering of pulverized chromosomes from micronuclei.

Cancer-associated genomic rearrangements from chromothripsis are accompanied by a characteristic DNA copy-number pattern that oscillates between two states, representing the retention and loss of fragments along the derivative chromosome[1,7–9]. Chromothripsis can be initiated by mitotic cell division errors resulting in the encapsulation of missegregated chromosomes into abnormal nuclear structures called micronuclei[2–5], which acquire extensive DNA damage in interphase upon nuclear envelope rupture[10–12]. After entry into mitosis, damaged chromosomes within micronuclei pulverize into dozens of microscopically visible fragments[13,14].

The stochastic inheritance of chromosome fragments by both newly formed daughter cells could in part contribute to the alternating DNA copy-number states that are characteristic of chromothripsis[2]. Sequencing of daughter cell pairs derived from micronucleated mother cells demonstrated that complex rearrangements are indeed a common outcome of micronucleus formation. However, in most cases, these patterns of chromothripsis differed from those in cancer genomes as the rearrangements were largely restricted to a single daughter cell and lacked the canonical oscillations in DNA copy-number states[2]. Moreover, germline chromothripsis events in congenital disorders

typically generate complex yet balanced rearrangements, indicative of minimal DNA loss[15,16]. These studies implicate a potential mechanism suppressing the loss of genetic material after chromosome pulverization, although how distinct patterns of rearrangements arise in cancer and germline disorders remains unclear.

The maintenance of a single centromere per chromosome is critical for establishing bipolar microtubule attachments to the mitotic spindle and achieving high-fidelity genome segregation[17]. However, most fragments derived from pulverized chromosomes are acentric and cannot directly bind to spindle microtubules[14]. Although several models have been proposed to promote acentric chromosome segregation[18–22], it is unclear how micronuclear fragments are inherited between daughter cells during mitosis—an important step for initiating its reassembly after reincorporation into the interphase nucleus[6]. Here we show that pulverized chromosomes from micronuclei spatially cluster throughout mitosis and identify the CIP2A–TOPBP1 complex as an essential regulator of this process. Mitotic clustering drives the unequal inheritance of acentric fragments by a single daughter cell, providing an explanation for the origins of distinct patterns of chromothripsis found across diverse cancer types and congenital disorders.

[1]Department of Pathology, University of Texas Southwestern Medical Center, Dallas, TX, USA. [2]European Molecular Biology Laboratory, European Bioinformatics Institute, Wellcome Genome Campus, Hinxton, UK. [3]Human Oncology and Pathogenesis Program, Department of Radiation Oncology, Memorial Sloan Kettering Cancer Center, New York, NY, USA. [4]Department of Cell Biology, Harold C. Simmons Comprehensive Cancer Center, University of Texas Southwestern Medical Center, Dallas, TX, USA. [5]Present address: Interdisciplinary Biomedical Graduate Program, University of Pittsburgh, Pittsburgh, PA, USA. [6]Present address: Department of Biochemistry & Cellular and Molecular Biology, University of Tennessee, Knoxville, TN, USA. [7]These authors contributed equally: Qing Hu, Alice Mazzagatti, Jose Espejo Valle-Inclán. ✉e-mail: icortes@ebi.ac.uk; peter.ly@utsouthwestern.edu

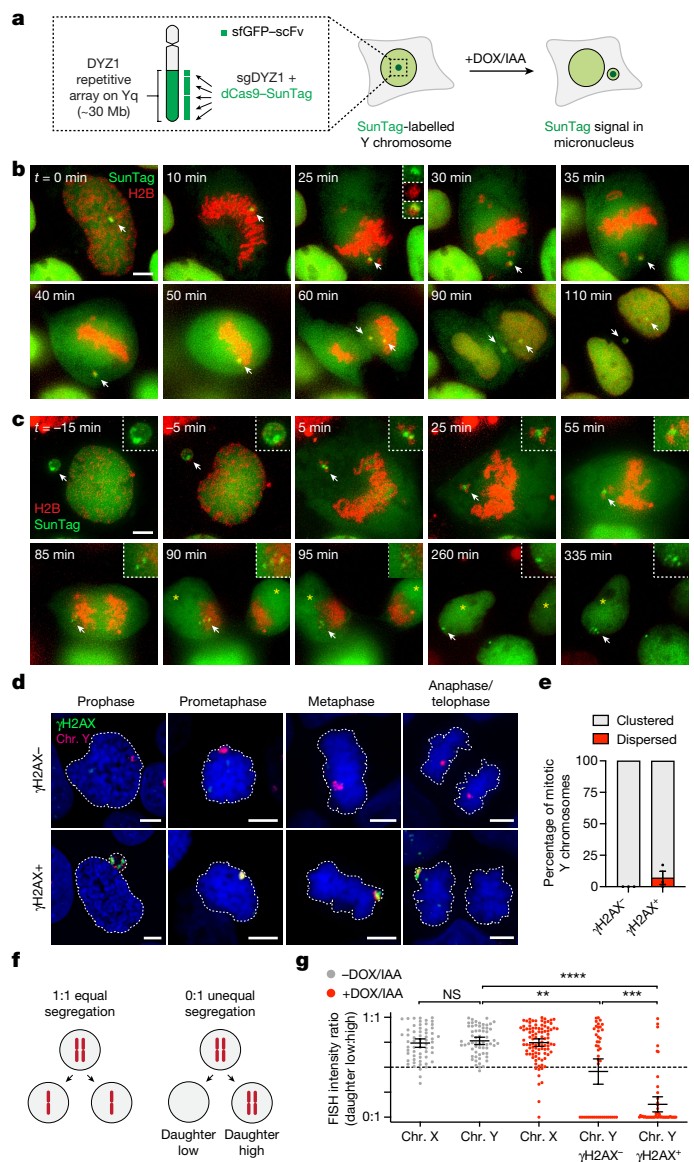

**a**, Schematic of the chromosome-labelling strategy using a dCas9–SunTag to target sfGFP fused to a single-chain variable fragment (sfGFP–scFv) to the Y-chromosome DYZ1 satellite array. Treatment with DOX/IAA triggers Y centromere inactivation and chromosome missegregation into micronuclei. **b**, Time-lapse images of a dCas9–SunTag-labelled Y chromosome (white arrow) missegregating into a micronucleus during mitosis after induction with DOX/IAA for 2 days. **c**, Time-lapse images of a dCas9–SunTag-labelled Y chromosome in a micronucleus clustering throughout mitosis with uneven distribution of the SunTag signal between daughter cells (yellow asterisks). A representative example is shown from $n = 13$ events obtained from independent experiments. In **b** and **c**, chromatin is labelled with H2B–mCherry. Scale bar, 5 μm. **d**, Micronuclear chromosome clustering in fixed DLD-1 cells at different stages of mitosis visualized by DNA FISH with chromosome paint probes. γH2AX was used to identify pulverized chromosomes from micronuclei with extensive DNA damage. DNA was stained with 4′,6-diamidino-2-phenylindole (DAPI) (blue). Scale bar, 5 μm. **e**, Quantification of fragment clustering with and without γH2AX from prometaphase to metaphase. Data are mean ± s.e.m. of $n = 3$ independent experiments; $n = 595$ (γH2AX⁻) and $n = 230$ (γH2AX⁺) cells. **f**, Schematic of chromosome distribution in a 1:1 or 0:1 segregation ratio between daughter cell pairs. **g**, FISH measurements between pairs of daughter cells indicate that pulverized chromosomes (chr.) are asymmetrically inherited by a single daughter. Data are mean ± 95% confidence intervals. Statistical analysis was performed using non-parametric Kruskal–Wallis test with correction for multiple comparisons; not significant (NS), $P > 0.05$; **$P = 0.0026$, ***$P = 0.0006$, ****$P ≤ 0.0001$. From left to right, $n = 61, 61, 95, 44$ and 51 daughter cell pairs pooled from 3 independent experiments. Representative images are shown in Extended Data Fig. 2f.

## Shattered chromosomes cluster in mitosis

We previously developed a model to induce the formation of micronuclei containing the Y chromosome in pseudodiploid human DLD-1 cells[3,14]. Exposure to doxycycline and auxin (DOX/IAA) triggers the replacement of the centromere-specific histone H3 variant CENP-A with a mutant that functionally inactivates the Y centromere without disrupting autosome or X chromosome segregation[14,23]. This system recapitulates the stepwise events of chromothripsis, including the pulverization of missegregated chromosomes in micronuclei that results in complex rearrangements[3,14]. To visualize the behaviour of micronucleated chromosomes during mitosis using live-cell imaging, we labelled the Y chromosome in DLD-1 cells using a nuclease-dead Cas9 (dCas9) fused to a SunTag scaffold (dCas9–SunTag; Methods), which can recruit 10–24 copies of superfolder green fluorescent protein (sfGFP) fused to a single-chain variable fragment[24]. As the Y-chromosome q arm comprises a ~30 megabase array containing DYZ1 repeats[25], we reasoned that a single sgRNA targeting DYZ1 could tile multiple dCas9–SunTag copies across half of the Y chromosome for labelling with sfGFP (Fig. 1a). To identify optimal CRISPR target sequences, we first generated stable cell lines encoding individual candidate sgRNAs, leading to the identification of a DYZ1 sgRNA that produced a single, nuclear sfGFP signal (Extended Data Fig. 1a,b). To reduce the heterogeneity in expression levels, clones derived from the parental population expressing

**Fig. 1 | Pulverized chromosomes from micronuclei spatially cluster throughout mitosis for biased inheritance by a single daughter cell.**

sfGFP under the control of various promoters or with different SunTag scaffold lengths were then screened for optimal, homogenous sfGFP levels (Extended Data Fig. 1c,d). As expected, dCas9–SunTag strongly co-localized with DNA fluorescence in situ hybridization (FISH) probes targeting the Y chromosome in interphase nuclei and on metaphase spreads (Extended Data Fig. 1e–g).

Using this chromosome-labelling strategy, we tested whether Y centromere inactivation could trigger the missegregation of the dCas9–SunTag-labelled Y chromosome. Treatment with DOX/IAA efficiently relocated dCas9–SunTag signals from the nucleus to micronuclei (Extended Data Fig. 1h,i). Time-lapse microscopy analysis confirmed frequent missegregation of the Y chromosome into micronuclei during mitosis (Fig. 1b and Supplementary Video 1). In the example shown, sister chromatid disjunction can be observed during anaphase by the resolution of a single SunTag puncta into two discrete signals, one of which missegregates into a micronucleus (Fig. 1b). To examine the fate of Y chromosome fragments in mitosis, cells with dCas9–SunTag-labelled micronuclei were first arrested in G2 phase using a CDK1 inhibitor. Most micronuclei had reduced background levels of diffused sfGFP, indicating that a proportion of observed events represented ruptured micronuclei with defective nucleocytoplasmic compartmentalization (Extended Data Fig. 1j)—a hallmark feature of micronuclei[10]. After release from G2, live-cell imaging revealed that micronuclear fragments unexpectedly remained clustered as a discrete dCas9–SunTag signal throughout mitosis in all of the events captured ($n = 13$; Fig. 1c and Supplementary Video 2). Notably, mitotic clustering resulted in the biased partitioning of most fragments into a single daughter nucleus (Fig. 1c), highlighting an unidentified mechanism that tethers chromosome fragments for biased partitioning as a collective unit.

To confirm these findings in fixed cells, we stained intact mitotic cells for γH2AX as a surrogate for micronuclear fragments with DNA double-stranded breaks (DSBs). After DOX/IAA treatment, nearly one-third of mitotic cells (31%, $n = 541$ cells) exhibited γH2AX signals that were specific for the Y chromosome. In agreement with live-cell dCas9–SunTag imaging, the majority (about 94%) of γH2AX-positive

Y-chromosome fragments were confined to a discrete region (Fig. 1d,e). Fragment clustering was observed throughout the entire duration of mitosis, indicating that they do not resolve at a specific mitotic stage and, indeed, the majority of signals were inherited by a single daughter cell after anaphase onset (Fig. 1d). Analyses of interphase nuclei at the corresponding timepoint revealed clusters of damaged chromosome fragments (Extended Data Fig. 2a–c) with expanded fluorescent signals (Extended Data Fig. 2d,e) that resembled subnuclear territories termed micronuclei bodies[26]. The nuclear envelope of micronuclei undergoes efficient disassembly during mitosis[3], suggesting that fragment clustering is not caused by failures in micronuclear envelope breakdown. Together, pulverized fragments from micronuclei have an intrinsic ability to cluster throughout mitotic cell division.

We further investigated the asymmetric inheritance patterns of fragmented chromosomes during mitosis by quantifying the intensity of Y-chromosome FISH probes between pairs of newly formed daughter cells (Fig. 1f). In unperturbed conditions, the Y chromosome and a control X chromosome segregated at an expected 1:1 ratio (Fig. 1g (grey points)). After Y centromere inactivation, 57% of intact Y chromosomes ($\gamma$H2AX-negative, $n = 44$ daughter pairs) segregated normally (ratio > 0.5:1), whereas 43% underwent whole-chromosome segregation errors resulting in the complete loss of the Y chromosome in one daughter cell (ratio = 0:1; Fig. 1g (red points)). By contrast, most (88%) pulverized Y chromosomes ($\gamma$H2AX-positive, $n = 51$ daughter pairs) were frequently inherited at an unequal ratio approaching 0:1, consistent with the segregation of most fragments into a single daughter nucleus (Fig. 1g). These data demonstrate that mitotic clustering of acentric fragments originating from micronuclei promotes the biased partitioning of pulverized chromosomes to one daughter cell.

## CIP2A–TOPBP1 mediates mitotic clustering

Several candidates of the DNA-damage response have been implicated in facilitating the segregation of acentric chromosomes and/or suppressing micronuclei formation, including DNA polymerase theta (encoded by *POLQ*) and the MRE11–RAD50–NBS1 complex in flies[27], as well as the CIP2A–TOPBP1 complex in mammalian cells[28–30]. To determine whether these components of the DNA-damage response are involved in fragment clustering, we assessed mitotic chromosome clustering in CRISPR–Cas9-edited *POLQ*[−/−], *NBN*[−/−] and *CIP2A*[−/−] DLD-1 clones (Fig. 2a). Whereas *POLQ* and *NBN*-knockout (KO) cells were similar to wild-type (WT) controls, cells lacking CIP2A exhibited a notable dispersion of Y-chromosome fragments (Fig. 2b) across independent clones generated with distinct sgRNAs targeting exon 1 (sg3 and sg4; Extended Data Fig. 3a).

*CIP2A*-KO cells proliferated at a slightly reduced rate (Extended Data Fig. 3b) but exhibited a normal cell cycle distribution profile (Extended Data Fig. 3c). As predicted, treatment with DOX/IAA triggered micronucleation and Y chromosome fragmentation at a frequency comparable to WT cells (Extended Data Fig. 3d,e), suggesting that CIP2A is not involved in driving whole-chromosome segregation errors or DNA-damage formation within micronuclei. However, mitotic *CIP2A*-KO cells contained Y-chromosome fragments co-localizing with $\gamma$H2AX that were noticeably displaced from the primary genomic mass (Fig. 2c). Fragment dispersion was evident in unsynchronized mitotic cells and those arrested with inhibitors that interfered with microtubule polymerization or the spindle assembly checkpoint (Extended Data Fig. 3f). Complementation of *CIP2A*-KO cells (generated by a frameshift deletion in exon 3) with full-length CIP2A fused to a HaloTag (CIP2A–HaloTag) fully rescued mitotic fragment clustering (Fig. 2d and Extended Data Fig. 3g).

Up to 45 distinct fragments were detected by microscopy in *CIP2A*-KO cells, whereas WT cells infrequently (13.6%, $n = 214$ cells) contained more than 5 dispersed fragments (Extended Data Fig. 4a). In WT cells, depletion of either CIP2A or its interacting partner TOPBP1 (refs. 28,30,31)

disrupted mitotic clustering (Extended Data Fig. 4b–d). Depletion of MDC1—which interacts with CIP2A–TOPBP1[28–30]—partially abolished clustering, although most cells continued to maintain fragmented chromosomes in proximity (Extended Data Fig. 4b–d). Time-lapse imaging showed that knockdown of CIP2A triggered the dispersion of chromosome fragments (Supplementary Video 3) and increased the number of detectable mitotic dCas9–SunTag-labelled signals in live cells (Extended Data Fig. 4e,f). Importantly, mitosis-specific degradation of CIP2A fused to a FKBP12(F36V) degron using the small-molecule degrader dTAGv-1 (ref. 32) in mitotically synchronized cells (Fig. 2e,f) was sufficient to disperse micronuclear fragments (Fig. 2g,h), indicating that fragment clustering is mediated by a mitotic function of CIP2A rather than a role during interphase, such as DNA repair.

To determine the spatial arrangement of micronuclear chromosome fragments, we analysed metaphase spreads prepared from mitotic DLD-1 cells swollen by hypotonic treatment. After the induction of micronucleation and chromosome fragmentation, DNA FISH using Y-chromosome painting probes revealed different degrees of fragment spreading, ranging from those that remained in close proximity to those that were scattered throughout the metaphase spread area (Extended Data Fig. 4g–i). Notably, both *CIP2A*-KO cells and WT cells depleted of CIP2A or TOPBP1 displayed a higher degree of metaphase chromosome fragment dispersion (Extended Data Fig. 4j,k).

Mitotic clustering was further confirmed in non-transformed RPE-1 cells containing chromosome 1 micronuclei induced non-randomly by nocodazole arrest[33] (Extended Data Fig. 5a–d). Loss of CIP2A triggered visibly dispersed chromosome 1 fragments that were nearly undetectable under control conditions (Extended Data Fig. 5e–g). Moreover, in renal proximal tubule epithelial cells (RPTECs), Cas9-mediated cleavage of chromosome 3p to produce micronuclei containing an acentric chromosome arm[34] (Extended Data Fig. 5h,i) showed similarly dispersed chromosome 3p fragments in *CIP2A*-KO cell populations (Extended Data Fig. 5j,k). Together, these efforts identify the CIP2A–TOPBP1 complex as an essential mitotic regulator involved in the spatial clustering of pulverized chromosomes from micronuclei.

## CIP2A–TOPBP1 in interphase micronuclei

To determine whether CIP2A associates with micronuclear fragments before mitotic entry, we assessed the interphase localization of CIP2A, which contains a nuclear export signal (NES) that drives its compartmentalization within the cytoplasm[30]. Consistent with this, CIP2A was rarely observed within intact micronuclei. However, using $\gamma$H2AX, accumulation of cGAS and/or loss of H3K9ac as markers of micronuclear envelope rupture[10,13,35,36], two distinct patterns of CIP2A emerged (Extended Data Fig. 6a,b). First, diffused CIP2A matching the intensity of the cytoplasmic pool was observed in one-third of ruptured micronuclei (Fig. 3a and Extended Data Fig. 6c,d), which is probably caused by defects in nucleo-cytoplasmic compartmentalization. Second, CIP2A appeared as robust puncta, which were less frequent (around 15%) but displayed a strong association with micronuclear envelope rupture (Fig. 3a and Extended Data Fig. 6c,d). Thus, whereas CIP2A does not associate with interphase DSBs in the nucleus[28,30,37], cytoplasmic CIP2A diffuses into ruptured micronuclei in which it prematurely engages with micronuclear DNA lesions that accumulate throughout interphase. In agreement, a mutant CIP2A rescue lacking its NES was sufficient to restore fragment clustering in *CIP2A*-KO cells (Fig. 2d and Extended Data Fig. 3g), suggesting that the normal cytoplasmic localization of CIP2A is dispensable for mitotic clustering following micronuclear envelope rupture.

TOPBP1 is normally nuclear localized and sequestered away from cytoplasmic CIP2A[28,30]. Intense TOPBP1 puncta were visible almost exclusively within ruptured, but not intact, micronuclei (Extended Data Fig. 6e–g). Notably, CIP2A and TOPBP1 formed highly co-localized puncta within micronuclei during interphase, as observed in multiple human cell lines containing micronuclei induced by distinct methods

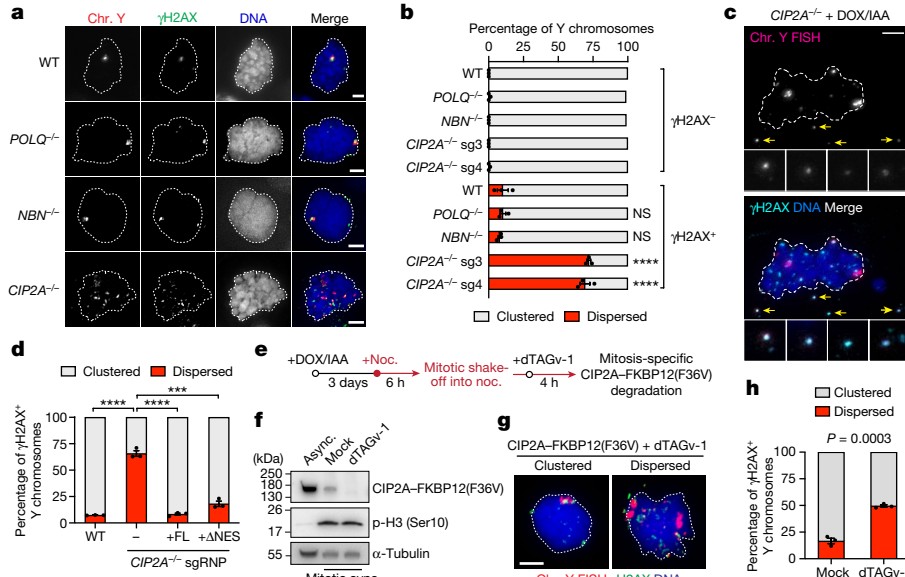

**Fig. 2 | Inactivation of CIP2A disrupts mitotic clustering and disperses pulverized chromosome fragments. a**, DOX/IAA-treated mitotic DLD-1 cells with the indicated genotypes with γH2AX-positive Y chromosomes. sg3 and sg4 denote distinct sgRNAs targeting exon 1 of *CIP2A*. **b**, Fragment clustering from γH2AX-negative and γH2AX-positive Y chromosomes from **a**. Data are mean ± s.e.m. Statistical analysis was performed using one-way ANOVA with correction for multiple comparisons compared with the WT controls; ****$P \leq 0.0001$. $n = 3$ independent experiments; from top to bottom, 657, 455, 352, 526, 383, 238, 165, 175, 179 and 149 cells. **c**, Mitotic *CIP2A*-KO (sg3) cell with γH2AX-positive Y-chromosome fragments displaced from the genomic mass (arrows). **d**, Fragment clustering from γH2AX-positive Y chromosomes from *CIP2A*-KO cells rescued by ectopic CIP2A–HaloTag. Data are mean ± s.e.m.

Statistical analysis was performed using two-tailed *t*-tests compared with *CIP2A*-KO cells; ****$P \leq 0.0001$, ***$P = 0.0002$. $n = 3$ independent experiments; 137 (WT), 200 (no rescue), 139 (full length (FL)) and 115 (ΔNES) cells. **e**, Schematic of inducing the depletion of CIP2A fused to a FKBP12(F36V) degron with dTAGv-1 in mitosis-synchronized cells. noc., nocodazole. **f**, Immunoblot analysis showing rapid degradation of CIP2A–FKBP12(F36V) after 4 h treatment of mitotic cells with dTAGv-1. Async, asynchronous; sync., synchronized. **g**, Clustered and dispersed Y-chromosome fragments after mitotic CIP2A depletion. **h**, Fragment clustering from γH2AX-positive Y chromosomes from **g**. Data are mean ± s.e.m. Significance was determined using two-tailed Student's *t*-tests. $n = 3$ independent experiments; 194 (mock) and 188 (dTAGv-1) cells. For **a**,**c** and **g**, scale bars, 5 μm.

(Fig. 3b,c). Consistent with an MDC1-independent function (Extended Data Fig. 4d), the loss of γH2AX—which directly recruits MDC1 to DSBs[38,39]—had no effect on CIP2A–TOPBP1 recruitment to ruptured micronuclei (lacking H3K9ac) in *H2AX*[−/−] RPE-1 cells that were treated with CENP-E/MPS1 inhibitors (Fig. 3b and Extended Data Fig. 6d,g). To visualize this process in live cells, we performed time-lapse imaging of *CIP2A*-KO cells reconstituted with CIP2A–HaloTag (Extended Data Fig. 3g), confirming the interphase puncta localization of CIP2A within ruptured micronuclei, as determined by the absence of sfGFP fused to a nuclear localization signal (NLS) (Fig. 3d). Importantly, as the nuclear envelope disassembled during mitotic entry, the micronucleated CIP2A–HaloTag puncta remained coalesced until the completion of mitosis, resulting in its partitioning exclusively to a single daughter cell (Fig. 3d and Supplementary Video 4). Thus, after the loss of nucleocytoplasmic compartmentalization within ruptured micronuclei, cytoplasmic CIP2A and nuclear TOPBP1 prematurely associate with interphase DNA lesions. This process occurs independently of MDC1, suggesting that the underlying DNA lesions sensed by CIP2A–TOPBP1 probably originate from incomplete DNA replication intermediates[28,40].

We next examined the localization of CIP2A–TOPBP1 on chromosome fragments during mitosis. Consistent with previous reports[28,30], CIP2A formed small co-localized foci with spontaneous mitotic DNA lesions in unperturbed cells (Extended Data Fig. 7a (top)). After DOX/IAA induction, a highly specific association between large CIP2A puncta with clusters of γH2AX-positive Y-chromosome fragments was observed in both mitotic cells (Extended Data Fig. 7a (bottom)) and pulverized metaphase chromosomes, but CIP2A puncta were undetectable on intact chromosomes (Fig. 3e,f). TOPBP1 was similarly recruited to clustered mitotic chromosome fragments in WT but not *CIP2A*-KO cells (Extended Data Fig. 7b–d), suggesting that TOPBP1 function during mitosis is

dependent on CIP2A, as shown for spontaneous mitotic DNA damage[30]. In agreement with MDC1-independent mitotic clustering, depletion of MDC1 reduced, but did not completely abrogate, the localization of both CIP2A and TOPBP1 to mitotic chromosome fragments (Extended Data Fig. 7e,f). Together, we propose that CIP2A–TOPBP1 bound to micronuclear DNA lesions poises acentric fragments for clustering immediately upon mitotic entry, which subsequently tethers pulverized chromosomes in close spatial proximity throughout mitosis (Fig. 3g).

To determine whether CIP2A–TOPBP1 interacts with acentric chromosomes in the absence of DNA lesions, we examined the mitotic localization of CIP2A and TOPBP1 on extrachromosomal DNA (ecDNA) elements, which are acentric by-products of chromothripsis[2,3,41] that nonetheless replicate normally and lack DSB ends due to their circularized nature. To do so, we used PC3 cells, which contain abundant ecDNAs in the form of double minute chromosomes that are visible on metaphase spreads, alongside ecDNA-negative HeLa S3 cells as a control (Extended Data Fig. 8a). In unperturbed mitotic PC3 and HeLa S3 cells, both CIP2A and TOPBP1 foci were not visible except for those that co-localized with spontaneous DSBs (Extended Data Fig. 8b,c). By contrast, DSBs induced by ionizing radiation stimulated the appearance of both mitotic CIP2A and TOPBP1 foci (Extended Data Fig. 8b,c). The CIP2A–TOPBP1 complex is therefore not normally recruited to ecDNAs during mitosis, indicating that its association with acentric chromosome fragments following chromothripsis requires the presence of abnormal DNA lesions.

## Genomic consequences of CIP2A loss

During mitosis, loss of CIP2A resulted in both daughter cells stochastically inheriting fragments of the pulverized chromosome (Fig. 4a,b).

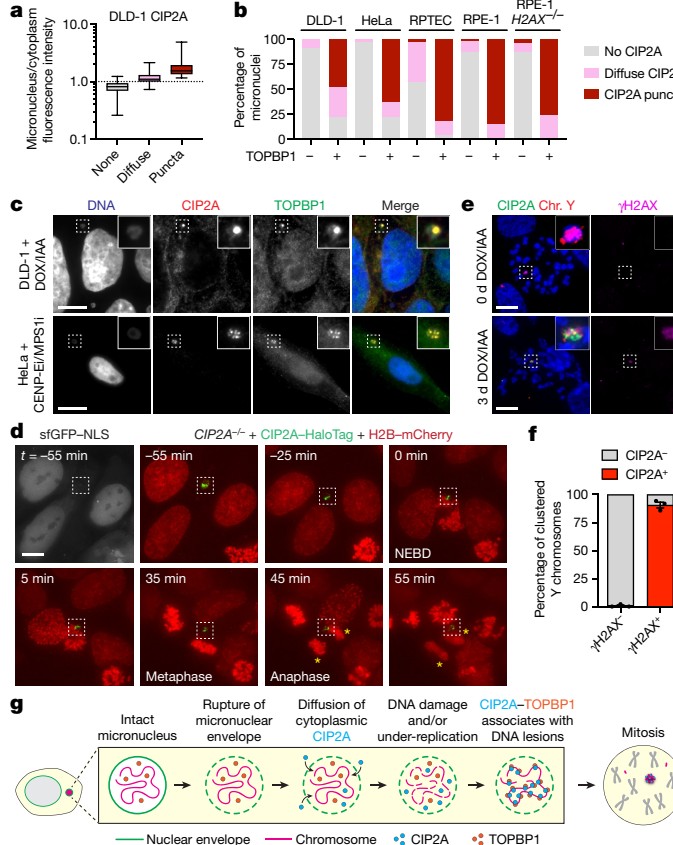

**Fig. 3 | Recruitment of CIP2A–TOPBP1 to ruptured micronuclei poises acentric fragments for clustering upon mitotic entry. a**, Intensity measurements of distinct CIP2A localization patterns in micronuclei compared with to the cytoplasmic pool. The box plots show the median (centre line) and the interquartile range (box limits) with the minimum–maximum values (whiskers). $n = 392$ (none), $n = 99$ (diffuse), $n = 23$ (puncta) micronuclei from 3 independent experiments. Example images are provided in Extended Data Fig. 6. **b**, The frequency of TOPBP1-positive micronuclei with the indicated patterns of CIP2A across a panel of human cell lines containing micronuclei induced by various methods. Data are mean. From left to right, $n = 165, 33, 182, 27, 290, 85, 208, 40, 247$ and $25$ micronuclei pooled from 2 (DLD-1 and HeLa) or 3 (RPTEC and RPE-1) independent experiments. **c**, Co-localization of CIP2A and TOPBP1 puncta in micronuclei of DLD-1 cells with Y-chromosome micronuclei (top) and HeLa cells with micronuclei containing random chromosomes (bottom). Scale bar, 10 μm. **d**, Time-lapse example of interphase CIP2A–HaloTag signal in a ruptured micronucleus (lacking sfGFP–NLS) through mitotic entry and the completion of mitosis. The yellow asterisks denote the two newly formed daughter cells. NEBD, nuclear envelope breakdown. Scale bar, 5 μm. **e**, Examples of mitotic chromosomes showing a highly specific association between CIP2A and clusters of γH2AX-positive Y-chromosome fragments (+DOX/IAA) but not γH2AX-negative Y chromosomes (−DOX/IAA). Scale bar, 10 μm. **f**, Quantification of CIP2A localization on Y chromosomes with and without γH2AX from **e**. Data are mean ± s.e.m. $n = 3$ independent experiments; 611 (γH2AX⁻) and 200 (γH2AX⁺) mitotic cells. **g**, Schematic of the stepwise series of events resulting in the premature engagement of CIP2A–TOPBP1 with DNA lesions following micronuclear envelope rupture during interphase.

After reincorporation into daughter cell genomes, multiple fragments were visibly dispersed throughout the nucleus, which subsequently recruited the DNA repair factor 53BP1 with similar efficiency to clustered micronuclei bodies (Extended Data Fig. 9a,b), consistent with engagement by the DNA damage response. We hypothesized that mutagenic repair of dispersed fragments distributed between both daughters would generate genomic abnormalities with a greater degree of chromosome loss. To directly test this, we examined 2,934 metaphase spreads for Y chromosomes with structural rearrangements using a dual-colour FISH approach[3] (Fig. 4c,d). Despite comparable fragmentation (Extended Data Fig. 3e) and rearrangement frequencies to WT cells (Fig. 4d), *CIP2A*-KO cells had less frequent complex rearrangements, as determined by the lack of co-localization of the two non-overlapping FISH probes (Fig. 4e). Selection for a Y-encoded neomycin-resistance marker[3] enabled the isolation of genetically stable Y chromosomes exhibiting complex rearrangements for further analysis (Fig. 4e), which were noticeably smaller in size in *CIP2A*-KO cells compared with those derived from WT controls (Fig. 4f). Thus, CIP2A-mediated mitotic clustering restricts the loss of genetic material on the pulverized chromosome.

In the absence of CIP2A, the small size and/or spatial positioning of dispersed mitotic fragments may pose a challenge for efficient reincorporation into daughter cell nuclei at the completion of mitosis (Fig. 4a (yellow arrows)). To visualize whether such fragments accumulated within the interphase cytoplasm, we stained semi-permeabilized DLD-1 cells with an antibody against double-stranded DNA (dsDNA), which enabled a focus on the cytoplasm while minimizing intense nuclear staining. Low levels of cytoplasmic DNA foci were triggered by micronucleation in WT cells, as we previously reported[42]. However, *CIP2A*-KO cells exhibited elevated baseline levels of cytoplasmic dsDNA foci that were exacerbated after micronucleus induction (Fig. 4g). Interphase FISH analysis of *CIP2A*-KO cells confirmed that cytoplasmic DNAs originated from the Y chromosome but not a control X chromosome (Fig. 4h,i). Notably, most cytoplasmic DNAs (79% from sg3 and sg4 combined, $n = 321$ *CIP2A*-KO cells) continued to contain active γH2AX marks (Fig. 4j) persisting from chromosomal damage accrued within micronuclei during the previous cell cycle.

At mitotic exit, the nuclear envelope reassembles around daughter cell genomes with the ability to also reform around individual chromosomes[43]. Small chromosome fragments positioned away from the genomic mass and/or mitotic spindle poles may be defective in establishing a proper nuclear membrane. To test this, we determined whether cytoplasmic DNAs comprised components of the nuclear membrane. Approximately half (57% and 56% in sg3 and sg4, respectively) of cytoplasmic DNA foci in *CIP2A*-KO cells contained detectable lamin B1 at an abundance comparable to the nucleus (Extended Data Fig. 9c,d). However, the remaining fraction of cytoplasmic DNAs were completely devoid of apparent lamin B1 (Extended Data Fig. 9e). Non-encapsulated cytoplasmic DNAs can activate the cGAS–STING pathway to trigger an innate immune response. Indeed, bulk RNA-sequencing (RNA-seq) analysis revealed activation of inflammatory responses in *CIP2A*-KO HeLa cells and, after induction of micronucleation with CENP-E/MPS1 inhibitors, this response was further accompanied by activation of the NF-κB pathway[42] and apoptosis-related transcriptional programs (Extended Data Fig. 9f). Thus, in addition to functioning as a source of cytoplasmic DNA after micronuclear envelope rupture[35,36,42], micronuclei can generate a second wave of cytoplasmic DNA owing to failures in reincorporating displaced chromosome fragments into daughter cell nuclei.

To determine the consequences of fragment dispersion on daughter cell fitness, we used extended live-cell imaging to track the fate of newly formed daughter cells generated by the division of micronucleated mother cells. Daughter cells lacking CIP2A were more susceptible to cell death during the subsequent interphase compared with control daughter cells (Fig. 4k). Despite mild proliferative defects under basal conditions (Extended Data Fig. 3b), *CIP2A*-KO cells were indeed sensitized to the induction of micronuclei in clonogenic assays (Extended Data Fig. 9g), probably due to activation of the DNA damage and/or immune response(s). We conclude that CIP2A inactivation renders cancer cells vulnerable to mitotic errors, which may represent a promising therapeutic target in combination with anti-mitotic agents or against chromosomally unstable or DNA-repair-deficient tumours.

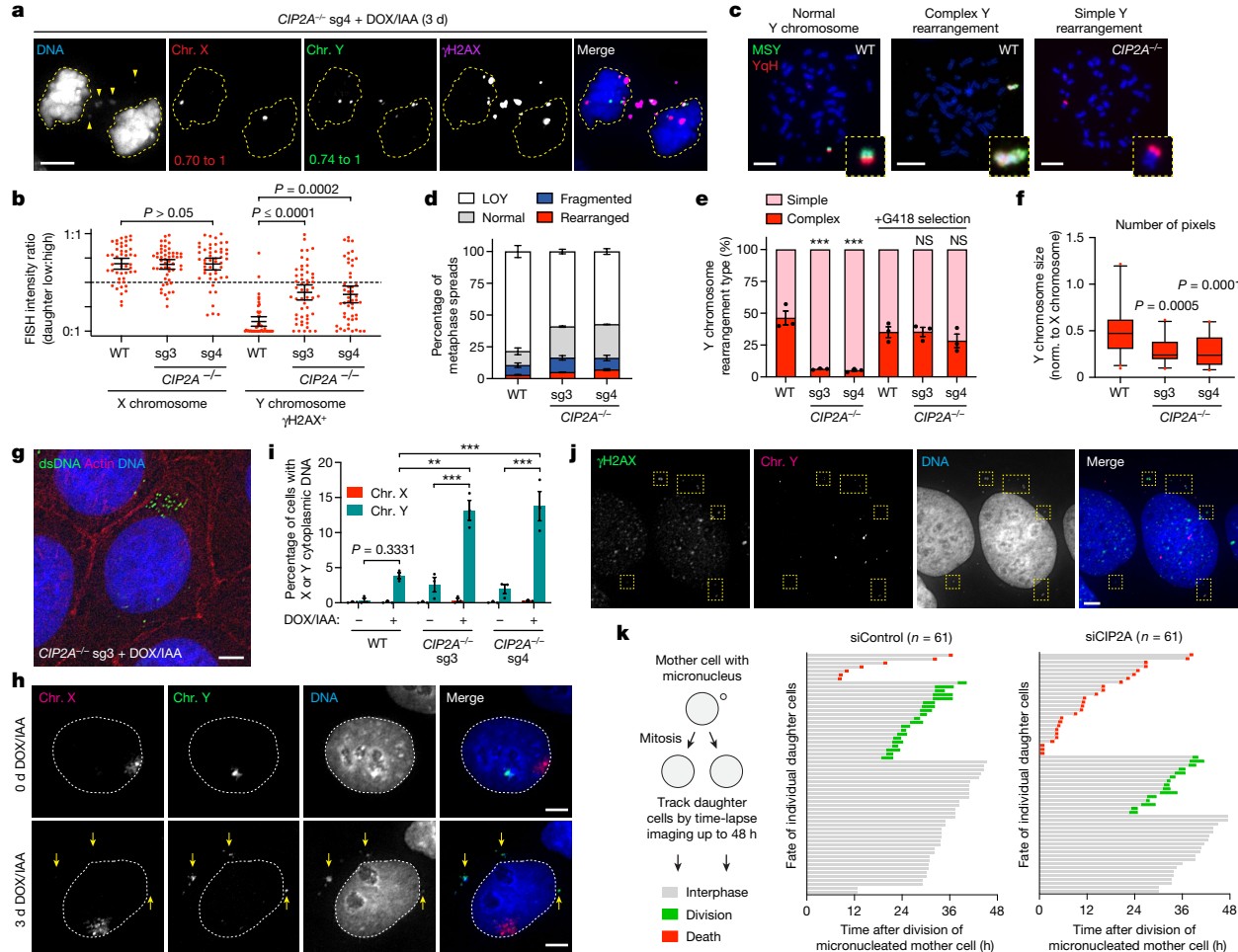

Fig. 4 | **Dispersed chromosome fragments randomly partition between daughter cells, undergo error-prone DNA repair and misaccumulate as cytoplasmic DNA. a**, CIP2A loss distributes pulverized chromosomes to both daughters during mitosis. **b**, FISH intensity ratios between daughter cell pairs. Data are mean ± 95% confidence intervals. *P* values were determined using non-parametric Kruskal–Wallis test with correction for multiple comparisons. *n* = 47 (WT), *n* = 51 (sg3) and *n* = 49 (sg4) daughter cell pairs pooled from 3 independent experiments. **c,d**, Examples (**c**) and frequencies (**d**) of Y-chromosome rearrangements using dual-coloured chromosome paint probes on WT and *CIP2A*-KO metaphases after DOX/IAA induction. Data are mean ± s.e.m. *n* = 3 independent experiments; 1,060 (WT), 990 (sg3) and 884 (sg4) metaphases. YqH, Y-chromosome q arm heterochromatic region. LOY, loss of the Y chromosome. **e**, Y-chromosome rearrangements with and without selection for a Y-encoded neomycin-resistance marker. Data are mean ± s.e.m. Statistical analysis was performed using one-way ANOVA; ***P = 0.0002. *n* = 3 independent experiments; from left to right, 29, 49, 62, 151, 166 and 156 metaphases. **f**, The number of DAPI pixels occupied by Y chromosomes with

complex rearrangements relative to the X chromosome. The box plots show the median (centre line) and 5th to 95th percentile (box limits). *P* values were determined using one-way ANOVA. *n* = 39 (WT), *n* = 23 (sg3) and *n* = 28 (sg4) chromosomes from 3 independent experiments. Norm., normalized. **g**, Semi-permeabilized DLD-1 cells stained with anti-dsDNA antibodies. Actin was used to demarcate cell boundaries. **h**, DNA FISH with chromosome paint probes showing Y-chromosome-specific cytoplasmic DNA foci in *CIP2A* sg3 KO cells. **i**, Quantification of the percentage of cells with X or Y chromosome-derived cytoplasmic DNA in **h**. Data are mean ± s.e.m. Statistical analysis was performed using one-way ANOVA with Tukey's multiple-comparisons test; **P = 0.0012, ***P ≤ 0.001. *n* = 3 independent experiments; from left to right, 2,410, 3,322, 1,960, 2,946, 2,100 and 3,940 cells. **j**, γH2AX-positive Y-chromosome fragments (yellow boxes) in the cytoplasm of interphase *CIP2A* sg4 KO cells. **k**, The fate of daughter cells from micronucleated mother cells was monitored over a 48 h period. Each bar represents a single daughter cell. Scale bars, 5 μm (**a**, **g**–**h** and **j**) and 10 μm (**c**).

## Balanced chromothripsis in cancer

Mitotic clustering biases the inheritance of shattered chromosome fragments from micronuclei towards a single daughter cell (Fig. 1c–g), thereby minimizing the loss of genetic material on the rearranged chromosome (Fig. 4a–f). In cancer genomes, this model predicts that a subset of chromothripsis cases would exhibit clusters of structural rearrangements lacking the DNA copy-number oscillations that are characteristic of canonical chromothripsis[1,8,44]. Evidence of such chromothripsis events—termed balanced chromothripsis—have been reported in the germline[45,46] and in lung and prostate adenocarcinomas[47,48]. However, the patterns, frequencies and consequences of balanced chromothripsis in cancer genomes remain largely

unclear owing to the commonly used requirement of detecting DNA copy-number oscillations to call chromothripsis events[8,44].

To examine this concept, we used ShatterSeek[8] to analyse whole-genome sequencing data from 2,575 tumours spanning 37 cancer types from the Pan-Cancer Analysis of Whole Genomes (PCAWG) Consortium for evidence of balanced chromothripsis (Methods). Applying a strict threshold for zero to minimal DNA copy-number changes, high-confidence balanced chromothripsis events were detected in around 5% of the cancer genomes analysed (119 of 2,575 tumour samples) (Fig. 5a). In the PCAWG cohort, the highest frequencies of balanced chromothripsis were found in prostate adenocarcinomas (19.6%), soft-tissue liposarcomas (15.8%) and bone osteosarcomas (11%). In prostate adenocarcinomas, these events were distinct from

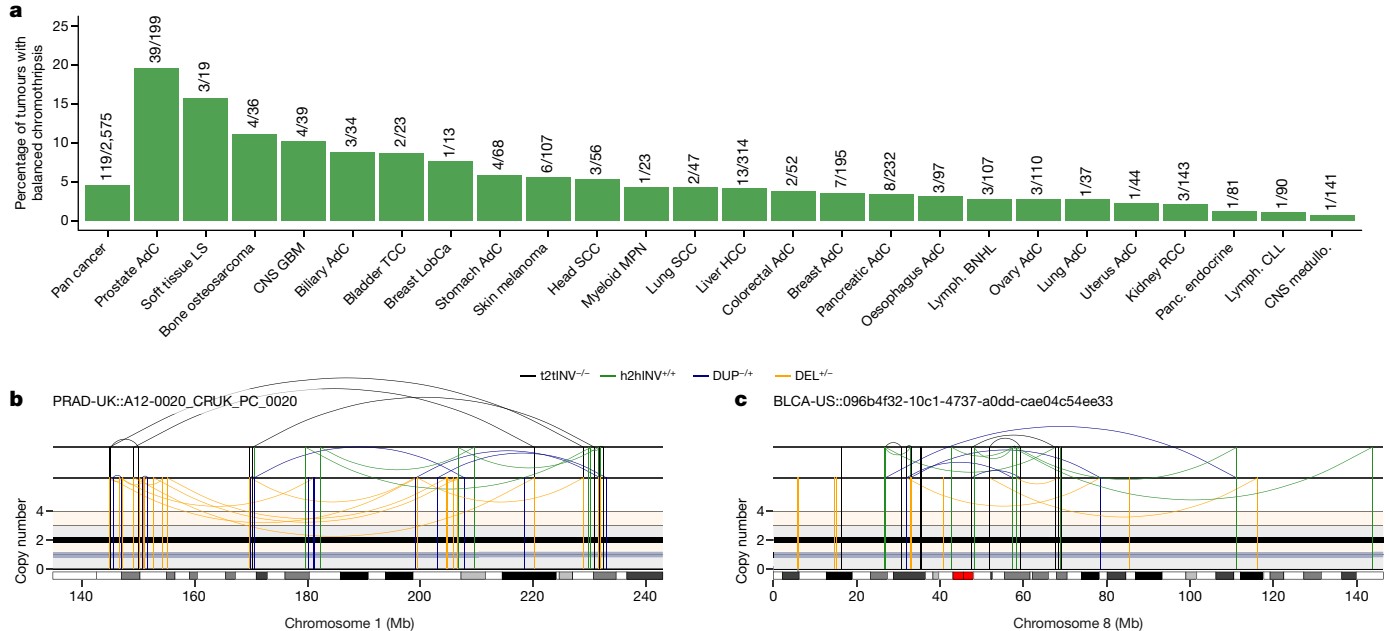

**Fig. 5 | Prevalence of DNA copy-number-neutral, balanced chromothripsis events across pan-cancer genomes. a**, The frequency of balanced chromothripsis in the ICGC/TCGA PCAWG cohort. The fractions represent the number of tumours with balanced chromothripsis in at least one chromosome over the total number of tumours of each type analysed. AdC, adenocarcinoma; CNS GBM, central nervous system glioblastoma; CNS medullo., central nervous system medulloblastoma; Head SCC, head-and-neck squamous cell carcinoma; HCC, hepatocellular carcinoma; LS, liposarcoma; LobCa, lobular carcinoma; Lymph. BNHL, lymphoid mature B cell lymphoma; Lymph. CLL, lymphoid chronic lymphocytic leukemia; MPN, myeloproliferative neoplasm; Panc. endocrine, pancreatic neuroendocrine tumour; RCC, renal cell carcinoma;

TCC, transitional cell carcinoma. **b**,**c**, Examples of balanced chromothripsis events in prostate adenocarcinoma (**b**) and bladder cancer (**c**) characterized by clusters of interleaved rearrangements, as expected for the random rejoining of genomic fragments shattered in chromothripsis, but without DNA loss, as indicated by the lack of deletions. The total and minor copy-number data are represented in black and grey, respectively. DEL, deletion-like rearrangement; DUP, duplication-like rearrangement; h2hINV, head-to-head inversion; t2tINV, tail-to-tail inversion. An example of canonical chromothripsis and additional examples of balanced chromothripsis events are provided in Extended Data Fig. 10.

chromoplexy as interchromosomal translocations were absent from the clusters of rearrangements. Balanced chromothripsis was found in more than one chromosome in 19 cases. Examples of canonical and balanced chromothripsis events are shown in Fig. 5b,c and Extended Data Fig. 10a-h, which exhibit complex and localized rearrangements reminiscent of chromothripsis but without the characteristic oscillations in DNA copy-number states. Among cancer genomes with a balanced chromothripsis event, 102 out of 119 (around 86%) disrupted at least one gene and 23 (about 19%) had chromosome breakpoints in putative cancer-driver genes, including the tumour suppressors *FOXO3*, *ARID2* and *PTEN* (Extended Data Fig. 10e,g; a complete list is provided in Supplementary Table 1). Moreover, balanced chromothripsis generated fusion genes in ten samples that included established oncogenic fusions, such as *CCDC170–ESR1* in breast adenocarcinomas, *RAB3C–PDE4D* in both skin melanomas and prostate adenocarcinomas, *CCT5–FAM173B* in bladder transitional cell carcinoma and *TMPRSS2–ERG* in prostate adenocarcinomas (Extended Data Fig. 10h and Supplementary Table 1). Together, these results show that balanced chromothripsis underpins the acquisition of cancer driver events across diverse tumour types.

## Discussion

We propose a multi-step model regulating the mitotic behaviour of acentric chromosome fragments from micronuclei (Extended Data Fig. 11). Rupture of the micronuclear envelope initiates the diffusion and mislocalization of CIP2A into micronuclei, where it can prematurely associate with TOPBP1 to engage with DNA lesions during interphase. Upon breakdown of the nuclear envelope at mitotic entry, the CIP2A–TOPBP1 complex facilitates the clustering of pulverized chromosomes

throughout mitosis. How CIP2A–TOPBP1 functions to tether fragments remains to be determined, but could occur through higher-order molecular interactions mediated by the extensive coiled-coil domain of CIP2A[28] and/or the condensate-forming property of TOPBP1 (ref. 49). Mitotic clustering promotes the biased partitioning of most chromosome fragments en masse to one daughter cell, which are then reincorporated into the interphase nucleus and manifest as micronuclei bodies[26] for engagement by error-prone DSB repair pathways. Finally, clustered fragments that are spatially positioned in nuclear proximity may become reassembled with increased DSB repair kinetics[50] to generate a spectrum of genomic rearrangements[3].

Mitotic clustering can safeguard against further genomic instability inflicted onto missegregated chromosomes; for example, by ensuring that most acentric fragments are inherited along with the centromere-containing fragment. Although this mechanism can minimize DNA copy-number loss, we propose several non-mutually exclusive explanations for the loss of genomic fragments associated with chromothripsis. First, some fragments may fail to participate in clustering owing to inefficiencies in tethering all acentric pieces. Additional factors may promote or inhibit CIP2A–TOPBP1 activity, and it remains unclear whether this regulation differs across cell and/or tissue types. Second, micronuclear DNAs exhibit under-replication during S phase[2,13], which can be caused by defective nucleocytoplasmic transport of the DNA replication machinery and/or the dilution of replication components following micronuclear envelope rupture. Finally, the loss of some fragments may arise from the inability of specific DSB repair pathways to reassemble all chromosome fragments, which then become lost throughout subsequent rounds of cell division.

The biased inheritance of acentric fragments by a single daughter may explain the origins of balanced chromothripsis, which were

detected in about 5% of pan-cancer genomes. This is likely a conservative estimate, as we applied a strict DNA copy-number loss threshold to limit detection to high-confidence samples that may represent one extreme of a spectrum. The prevalence of canonical chromothripsis in cancer genomes probably reflects strong positive selection pressure owing to the increased risk of tumour suppressor deletions caused by partial chromosome loss[1]. By contrast, balanced chromothripsis—which relies on the precise location of rearrangement breakpoints to disrupt gene(s)—may be better tolerated in the germline and more capable of generating karyotypes that are compatible with organismal development[15]. These factors may in part contribute to the complex yet balanced rearrangement landscapes that are found in congenital disorders[16]. Further studies are needed to define the specific cellular contexts in which these mechanisms are operative, as well as their contributions to cancer genome evolution and germline disorders.

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

# Methods

## Cell lines and reagents

All of the cell lines were maintained at 37 °C under 5% $CO_2$ and atmospheric oxygen. DLD-1, HeLa, PC3 (a gift from S. Wu), HEK293T and 293GP cells were cultured in Dulbecco's modified Eagle medium (DMEM, Gibco) supplemented with 10% tetracycline-free fetal bovine serum (Omega Scientific) and 100 U ml$^{-1}$ penicillin–streptomycin. RPE-1 and RPE-1 $H2AX^{-/-}$ cells (a gift from S. Jackson) were cultured in DMEM/F-12 (Gibco) supplemented with 10% fetal bovine serum and 100 U ml$^{-1}$ penicillin–streptomycin. RPTECs (a gift from D. Marciano) expressing a short hairpin RNA against *TP53* were cultured in renal epithelial cell growth basal medium (Lonza) supplemented with 0.5% tetracycline-free fetal bovine serum (Omega Scientific), 100 U ml$^{-1}$ penicillin–streptomycin, 10 ng ml$^{-1}$ human recombinant epidermal growth factor, 10 μg ml$^{-1}$ human transferrin, 1 μg ml$^{-1}$ hydrocortisone, 10 μM adrenaline, 50 ng ml$^{-1}$ triiodo-L-thyronine, 5 μg ml$^{-1}$ insulin, 30 μg ml$^{-1}$ gentamicin and 15 ng ml$^{-1}$ amphotericin B. Cell lines were authenticated by karyotyping and were routinely confirmed to be free of mycoplasma contamination using the Universal Mycoplasma Detection Kit (ATCC).

DOX and IAA (Millipore-Sigma) were dissolved in cell-culture-grade water and used at 1 μg ml$^{-1}$ and 500 μM, respectively, in DLD-1 cells. Geneticin (G418 sulfate) and zeocin (InvivoGen) were used at selection concentrations of 300 and 50 μg ml$^{-1}$, respectively. For cell-cycle-arrest experiments, 100 ng ml$^{-1}$ nocodazole (Millipore-Sigma), 100 ng ml$^{-1}$ Colcemid (KaryoMAX, Thermo Fisher Scientific) or 10 μM MG132 (a gift from J. Seemann) was used for mitotic arrest, and 10 μM of the CDK1 inhibitor RO-3306 (Millipore-Sigma) was used for G2 arrest, all of which were dissolved in dimethylsulfoxide. A total of 50 nM CENP-E inhibitor (GSK-923295, Cayman Chemical) and 480 nM MPS1 inhibitor (NMS-P715, Cayman Chemical) was used to induce chromosome segregation errors and micronuclei formation in HeLa and RPE-1 cells. dTAGv-1 (500 nM; a gift from B. Nabet) was used to induce degradation of FKBP12(F36V) fusion proteins. For ionizing-radiation experiments, cells were irradiated with γ-ray (2 Gy) generated by a Mark I $^{137}$Cs irradiator (JL Shepherd) and fixed 1 h after irradiation for immunofluorescence analysis. Small interfering RNA (siRNA) transfections were conducted with Lipofectamine RNAiMAX reagent (Thermo Fisher Scientific). siRNAs were synthesized (Thermo Fisher Scientific) and used at a final concentration of 20 nM. A list of all of the siRNA sequences used in this study is provided in Supplementary Table 2.

## Cell line engineering

To generate *CIP2A*-KO cells, target sequences for guide RNAs were designed using CRISPick (Broad Institute). Oligonucleotides encoding guide RNAs targeting exon 1 of *CIP2A* (sg3 and sg4; Supplementary Table 2) were cloned into the BsmBI restriction site of the Lenti-Cas9-gRNA-TagBFP2 vector (Addgene, 124774) and packaged in HEK293T cells by co-transfection with pMD2.G (Addgene 12259) and psPAX2 (Addgene, 12260) using X-tremeGENE 9 (Millipore-Sigma). Viral supernatants after 48 h or 72 h transfection were filtered (0.45 μm), and cells were infected in the presence of 5 μg ml$^{-1}$ polybrene (Millipore-Sigma) for around 24 h. Fluorescent cells were isolated using fluorescence-activated cell sorting (FACS) into 96-well plates (BD FACSAria II). For RPTECs (*CIP2A* sg3 and sg4) and HeLa (*CIP2A* sg1 and sg2) cells, KO populations were established by pooling together virus-infected cells. To establish KO clones, DLD-1 cells were sorted, expanded and verified by both Sanger sequencing and immunoblotting.

To generate DLD-1 KO cells by ribonucleoprotein (RNP)-mediated CRISPR genome editing, two sgRNAs per gene were synthesized (Synthego) and co-transfected with TrueCut Cas9 protein v2 (Invitrogen) using the Lipofectamine CRISPRMAX Cas9 transfection reagent (Invitrogen). A list of all of the sgRNA sequences used in this study is provided in Supplementary Table 2. After transfection, cells were plated by limiting dilution into 96-well plates. Single-cell-derived clones were expanded, screened by PCR for targeted deletions and confirmed to harbour frameshift deletion mutations by Sanger sequencing. A list of all PCR primers (Millipore-Sigma) is provided in Supplementary Table 2.

For complementation experiments, *CIP2A* cDNA (a gift from Q. Zhang) and HaloTag (Addgene, 112852) were cloned into pBABE-zeo (Addgene, 1766) and packaged in 293GP cells by co-transfection with pVSV-G using X-tremeGENE 9. *CIP2A*-KO cells generated by RNP-mediated gene editing were infected with retroviruses encoding full-length CIP2A, delta NES mutant (lacking amino acids 561–625) or CIP2A–FKBP12(F56V) fused to an N terminus HaloTag for 24 h and selected with zeocin for 10 days. For the expression of other exogenous genes, the H2B-mCherry (a gift from H. Yu) and cGAS-GFP constructs (a gift from Z. Chen) were used to generate viruses for transduction of DLD-1 cells, as described above.

## Chromosome labelling using dCas9–SunTag

To label the Y chromosome in live cells, the SunTag labelling system was adopted and modified as described below. DYZ1 repeats (3,584 bp, sequence information provided by H. Skaletsky) were analysed by CRISPick (Broad Institute) and five sgRNA sequences were selected for targeting DYZ1 repeats. scFv-GCN4-sfGFP-GB1-NLS from SunTag plasmid (Addgene, 60906) was cloned into a lentiGuide-puro vector (Addgene, 52963). Lentiviral supernatants, which were packaged in HEK293T cells by co-transfection with pMD2.G and psPAX2 with either lentiGuide-scFv-GCN4-sfGFP-GB1-NLS or pHRdSV40-dCas9-10xGCN4_v4-P2A-BFP (SunTag plasmid, Addgene, 60903), and retroviral supernatants, which were packaged in 293GP cells by co-transfection of pBABE-H2B-mCherry with pVSV-G after 48 h or 72 h transfection, were filtered (0.45 μm) and DLD-1 cells were infected in the presence of 5 μg ml$^{-1}$ polybrene (Millipore-Sigma) for around 24 h. Fluorescent cells were isolated by FACS (BD FACSAria II) and plated by limiting dilution into 96-well plates. Single-cell-derived clones were expanded and screened for expected SunTag signals.

## Live-cell imaging

DLD-1 cells expressing dCas9–SunTag and H2B–mCherry were plated into Nunc Lab-Tek chambered cover glasses. Images were acquired on the DeltaVision Ultra microscope (Cytiva) in a humidity- and temperature-controlled (37 °C) environment supplied with 5% $CO_2$ at 5 min intervals for 16 h using a ×60 objective with 11 × 0.5 μm $z$-sections under low power exposure. For CIP2A–HaloTag imaging, cells were labelled with 200 nM JF646 ligand (Promega) for 15 min and washed with fresh medium before imaging. Images were deconvolved and maximum-intensity quick projections were generated using softWoRx (v.7.2.1, Cytiva), and videos were analysed using Fiji (v.2.1.0/1.53c).

For long-term live-cell imaging, DLD-1 cells expressing H2B-mCherry were transfected with siRNAs and seeded in 96-well glass-bottom plates (Cellvis, P96-1.5H-N). The next day, cells were treated with DOX/IAA for 72 h. One day before image acquisition, cells were retransfected with siRNAs to ensure depletion of the target protein throughout the duration of the experiment. Images were acquired on the ImageXpress Confocal HT.ai High-Content Imaging System (Molecular Devices) in a humidity- and temperature-controlled (37 °C) environment in $CO_2$-independent medium at 15 min intervals for 48 h using a ×40 objective with 7 × 1.5 μm $z$-sections under low-power exposure. Maximum-intensity projections were generated using MetaXpress (Molecular Devices), and videos were analysed using Fiji (v.2.1.0/1.53c).

## Immunofluorescence analysis

DLD-1 cells were plated onto CultureWell gaskets (Grace Bio-Labs) and assembled glass slides were fixed with 4% formaldehyde for 10 min. For dispersion analysis, cells were arrested in mitosis for 4 h using Colcemid and collected by shake-off. Cell suspensions were concentrated to 1 × 10$^6$ cells per ml in PBS and centrifuged onto glass slides using

a Cytospin 4 cytocentrifuge (Thermo Fisher Scientific). Fixed cells were permeabilized with 0.3% Triton X-100 in PBS for 5 min, incubated with Triton Block (0.2 M glycine, 2.5% fetal bovine serum, 0.1% Triton X-100, PBS) and then incubated with primary antibodies. The following primary antibodies were used at the indicated dilutions in Triton Block: 1:500 anti-CIP2A (sc-80659, Santa Cruz), 1:500 anti-TOPBP1 (sc-271043, Santa Cruz), 1:300 anti-TOPBP1 (ABE1463, Millipore), 1:1,000 anti-phosphorylated H2AX (Ser139) (05-636, Millipore), 1:1,000 anti-phosphorylated H2AX (Ser139) (2577, Cell Signaling), 1:1,000 anti-53BP1 (NB100-304, Novus) antibodies. Cells were washed with 0.1% Triton X-100 in PBS, incubated with 1:1,000 dilutions of Alexa Fluor-conjugated donkey anti-rabbit or donkey anti-mouse secondary antibodies (Invitrogen) for 1 h at room temperature, and washed with 0.1% Triton X-100 in PBS. Immunostained cells were fixed with Carnoy's fixative for 15 min and rinsed with 80% ethanol. Air-dried cells were then used for DNA FISH, as described below.

For micronuclei analysis, DLD-1, HeLa and RPE-1 cells were grown on glass coverslips and fixed with PTEMF (0.2% Triton X-100, 0.02 M PIPES pH 6.8, 0.01 M EGTA, 1 mM $MgCl_2$ and 4% formaldehyde) for 10 min, followed by two washes in 1× PBS. The samples were blocked with 3% bovine serum albumin diluted in PBS. Cells were incubated for 1 h at room temperature with the following primary antibodies diluted in 3% BSA: 1:500 anti-CIP2A (sc-80659, Santa Cruz), 1:1,000 anti-CIP2A (14805, Cell Signaling), 1:500 anti-TOPBP1 (sc-271043, Santa Cruz), 1:500 anti-TOPBP1 (ABE1463, Millipore), 1:1,000 anti-phosphorylated histone H2AX (Ser139) (2577, Cell Signaling), 1:1,000 anti-acetyl-histone H3 (Lys9) (9649, Cell Signaling) and 1:1,000 anti-CGAS (15102, Cell Signaling). After three 5 min washes, Alexa-Fluor-conjugated donkey anti-rabbit or donkey anti-mouse secondary antibodies (Invitrogen) were diluted 1:1,000 in 3% BSA and applied to cells for 1 h at room temperature, followed by two 5 min washes with 1× PBS. DNA was counterstained with DAPI and cells were mounted in ProLong Gold antifade mounting solution.

For cytosolic dsDNA staining, cells were fixed with 4% formaldehyde for 10 min and then treated with 0.02% saponin in PBS for 5 min. Semi-permeabilized cells were incubated with blocking solution (2.5% fetal bovine serum in PBS) followed by incubation with anti-dsDNA antibodies (1:250 in blocking solution, sc-58749, Santa Cruz) at 4 °C overnight. After washing with PBS, cells were incubated with 1:1,000 dilutions of an Alexa-Fluor-conjugated donkey anti-mouse secondary antibody (Invitrogen) in blocking solution for 1 h and washed with PBS. Cells were then fully permeabilized with 0.3% Trion X-100 in PBS for 5 min and washed with PBS. Permeabilized cells were incubated with 5 U ml$^{-1}$ of fluorescent phalloidin (Biotium) in PBS for 20 min and washed with PBS.

## Metaphase spread preparation

Cells were treated with 100 ng ml$^{-1}$ Colcemid (KaryoMAX, Thermo Fisher Scientific) for 4–5 h before collection by trypsinization and centrifugation. Cell pellets were resuspended in 500 µl PBS followed by adding 5 ml of 75 mM KCl solution dropwise while gently vortexing. Cells were incubated for 6 min in 37 °C water bath and fixed using freshly prepared, ice-cold Carnoy's fixative (3:1 methanol:acetic acid), followed by centrifugation and resuspension in Carnoy's. Cells were subsequently dropped onto slides and air dried for further processing.

## DNA FISH

DNA FISH probes (MetaSystems) were applied to metaphase spreads and sealed with a coverslip using rubber cement. Slides were co-denatured on a heat block at 75 °C for 2 min and then hybridized at 37 °C in a humidified chamber overnight. The next day, the coverslips were removed, and the slides were washed with 0.4× SSC at 72 °C for 2 min and rinsed with 2× SSC with 0.05% Tween-20 at room temperature for 30 s. After washing, the slides were counterstained with DAPI, air dried and mounted in ProLong Gold antifade mounting solution.

## Fixed-cell microscopy

Immunofluorescence images were captured on a DeltaVision Ultra (Cytiva) microscope system equipped with a 4.2 Mpx sCMOS detector. Interphase nuclei and micronuclei images were acquired with a ×100 objective (UPlanSApo, 1.4 NA) and 1 × 0.2 µm z-section. Quantitative fluorescence image analyses were performed using Fiji (v.2.1.0/1.53c). IF−FISH images were acquired with a ×60 objective (PlanApo N 1.42 oil) and 15 × 0.2 µm z-sections. Deconvolved maximum intensity projections were generated using softWoRx (v.7.2.1, Cytiva).

Metaphase FISH images were acquired on the Metafer Scanning and Imaging Platform microscope (Metafer 4, v.3.13.6, MetaSystems). The slides were first scanned for metaphases using M-search with a ×10 objective (ZEISS Plan-Apochromat 10x/0.45), and metaphases were automatically imaged using Auto-cap with a ×63 objective (ZEISS Plan-Apochromat 63x/1.40 oil). Images were analysed using the Isis Fluorescence Imaging Platform (MetaSystems) and Fiji (v.2.1.0/1.53c).

## Chromosome distribution between daughter cells

DLD-1 cells were seeded in four-well chamber slides and treated with or without DOX/IAA for 48 h. Cells were then arrested in G2 with 10 µM CDK1 inhibitor RO-3306 (Millipore-Sigma) for 16 h, washed with PBS three times and released into mitosis in fresh medium. After 90 min, cells were fixed with 4% formaldehyde followed by IF−FISH, as described above, and hybridized to X- and Y-chromosome paint probes (MetaSystems). For analysis of chromosome inheritance between daughter cells, pairs of newly formed daughter cells were imaged on the DeltaVision Ultra (Cytiva) microscope system. Images were split into separate channels for quantification using the ImageJ plugin Segmentation (Robust Automatic Threshold Selection) to create a mask for the FISH signals. Particles of the mask were analysed to generate a list of regions of interest for intensity measurements. FISH signal intensities were then measured in each pair of daughter cells for both the X and Y chromosomes. The distribution of FISH signal was calculated by the ratio of the daughter cell with the lower signal compared to the daughter cell with the higher signal.

## Mitosis-specific depletion of FKBP fusion proteins

*CIP2A*-KO DLD-1 cells complemented with CIP2A-FKBP12(F36V) were seeded in T75 flasks and treated with DOX/IAA for 72 h. Cells were than arrested in mitosis with 100 ng ml$^{-1}$ nocodazole for 6 h and mitotic cells were collected by mitotic shake-off. Mitotic cells were then reseeded in 24-well plates and treated with or without 500 nM dTAGv-1 for 4 h in the presence of 100 ng ml$^{-1}$ nocodazole. Cells were centrifuged onto glass slides using the Cytospin 4 cytocentrifuge and processed for IF−FISH.

## Chromosome fragment dispersion

Metaphase spreads were prepared as described and hybridized to Y-chromosome paint probes (MetaSystems). Metaphases with fragmented Y chromosomes were identified and split into separate channels. Fragment dispersion was analysed using the ImageJ plugin HullAndCircle to measure the convex hull of the Y-chromosome fragments relative to all DAPI-stained chromosomes. Dispersion indices were calculated by dividing the area of Y-chromosome fragments by the overall DAPI area followed by minimum−maximum normalization of all data points within each sample.

## Chromosome-specific micronuclei in RPE-1 cells and RPTECs

RPE-1 cells were seeded in T175 flasks and transfected with siRNAs the next day. One day after transfection, cells were arrested in mitosis with 100 ng ml$^{-1}$ nocodazole for 8 h. Mitotic cells were collected by mitotic-shake off, washed three times with culture medium and reseeded onto coverslips and T75 flasks. Cells growing on coverslips were fixed at 20 h after releasing from mitosis for analysis of chromosome 1 micronuclei by FISH. For dispersion analysis, after 20 h release

from nocodazole, cells growing on T75 flasks were arrested in mitosis with Colcemid for 4 h. Mitotic cells were collected by mitotic-shake off, centrifuged onto glass slides using the Cytospin 4 cytocentrifuge and processed for FISH.

RPTECs were seeded in T75 flasks and transfected with Cas9 (TrueCut Cas9 protein v2, Thermo Fisher Scientific) in a complex with an sgRNA targeting chromosome 3p using Lipofectamine CRISPRMAX Cas9 transfection reagent (Thermo Fisher Scientific). During transfection, 3 μM of the DNA-PK inhibitor AZD7648 (MedChemExpress) was added for 24 h and washed out. Three days later, cells were arrested in mitosis with Colcemid for 6 h. Mitotic cells were collected by mitotic-shake off, centrifuged onto glass slides using the Cytospin 4 cytocentrifuge and processed for FISH.

### Cell cycle profiling
Cells were trypsinized, washed with PBS and fixed with 70% ethanol in PBS at −20 °C for 2 h. Fixed cells were washed with PBS twice and incubated with staining solution (0.1 mg ml$^{-1}$ RNase A, 0.1% Triton X-100, 10 μg ml$^{-1}$ propidium iodide). Cells were analysed using a FACSCalibur (BD Biosciences) flow cytometer, and cell cycle profiles were generated using FlowJo (v.10.8.2, BD Biosciences) software.

### Immunoblotting
Whole-cell extracts were collected in Laemmli SDS sample buffer and boiled for 5 min. The samples were resolved by SDS polyacrylamide gel electrophoresis, transferred to polyvinylidene fluoride membranes and blocked with 5% milk diluted in PBST (PBS, 0.1% Tween-20). The following primary antibodies were diluted in PBST and used: 1:1,000 anti-CIP2A (sc-80659, Santa Cruz), 1:5,000 anti-α-tubulin (3873, Cell Signaling), 1:1,000 anti-TOPBP1 (sc-271043, Santa Cruz), 1:1,000 anti-phosphorylated histone H3 (Ser10) (06-570, Millipore) and 1:5,000 anti-MDC1 (ab11171, Abcam). The blots were incubated with 1:4,000 dilutions of horseradish peroxidase-conjugated goat anti-rabbit or donkey anti-mouse secondary antibodies (Invitrogen), incubated with SuperSignal West Pico Plus chemiluminescent substrate (Thermo Fisher Scientific) and processed using the ChemiDoc MP imaging system (Bio-Rad).

### Cell proliferation assays
To measure cell proliferation, $1 \times 10^5$ cells were seeded onto p60 mm dishes in triplicate, treated with or without DOX/IAA the next day and counted at three-day intervals. For clonogenic survival assays, 1,000 cells were plated in p60 mm dishes in triplicate for 15 days. Colonies were fixed in ethanol, stained with 0.5% crystal violet/70% ethanol solution and manually counted.

### Analysis of Y-chromosome rearrangements
Two-colour DNA FISH probes (MetaSystems) were applied to metaphase spreads and captured on the Metafer Scanning and Imaging Platform (Metafer 4, v.3.13.6, MetaSystems), as described above. Distinct types of structural rearrangements were manually inspected using previously described criteria[3,51]. To determine the size of Y chromosomes with complex rearrangements, images were split into separate channels using Fiji (v.2.1.0/1.53c) followed by creation of a mask by segmentation of the DAPI channel using Threshold adjust. Particles of the mask were analysed to generate regions of interest for area measurement. The number of DAPI-occupied pixels of rearranged Y chromosomes were measured and normalized to the X chromosome from the same metaphase spread.

### RNA-seq analysis
HeLa cells were transduced with a control sgRNA (sgNTC), sgCIP2A-1 or sgCIP2A-2 and selected with puromycin. Total RNA from three independent biological replicates was collected using the RNeasy Total RNA kit (Qiagen), and libraries were sequenced on the Illumina NovaSeq

6000 platform (Novogene). Sequencing reads were aligned to the transcriptome using STAR (v.2.7.4a)[52]. Gene expression counts were generated using HTSeq (v.0.6.1p1)[53] and normalized to transcripts per kilobase million. GENCODE (v.22) was used as the gene annotation reference[54]. Gene Set Enrichment Analysis (GSEA, v.4.3.2)[55] was performed using the weighted enrichment statistic on normalized gene counts computed using DESeq2[56]. We used Hallmark gene sets containing between 15 and 500 genes from the Human Molecular Signatures Database (MSigDB)[57].

### Whole-genome sequencing analyses
To detect copy-number-balanced chromothripsis events, we applied ShatterSeek[8] (v.1.1; https://github.com/parklab/ShatterSeek) to 2,575 tumour–normal pairs from PCAWG that passed quality-control criteria. We considered all chromosomes with a cluster of at least five structural variants (SVs). We considered all clusters irrespective of the number of copy-number oscillations in the cluster. To call a cluster of SVs a copy-number-balanced chromothripsis event, we required: (1) at least five intrachromosomal SVs; (2) no translocation mapping to the genomic region encompassed by the cluster of SVs; we included this filter to distinguish balanced chromothripsis from chromoplexy events, which are characterized by chains of interchromosomal SVs with limited genomic DNA loss and could therefore be misclassified as balanced chromothripsis if this filter was not applied; (3) no overlap with chromoplexy calls generated for these tumours using ChainFinder[58] as previously reported[8]; and (4) that less than 1% of the genomic region encompassed by the cluster of SVs shows a copy number of less than the modal copy number of the chromosome. We applied this filter to ensure that balanced chromothripsis calls do not contain canonical chromothripsis events. Finally, all cases that passed these filters were examined manually by visualizing genomic rearrangement plots using ReConPlot[59].

To find gene disruptions within the balanced chromothripsis clusters, we first downloaded gene coordinates from Ensembl[60] (GRCh37) using biomaRt[61]. We next intersected the coordinates of the breakpoints and genes using bedtools[62]. We considered a gene to be disrupted if a breakpoint mapped within the region defined by the start and end coordinates of the gene ±5 kilobases. We determined putative cancer-driver genes using the pan-cancer driver catalogue from the Hartwig Medical Foundation cancer whole-genome sequencing analysis pipeline (https://github.com/hartwigmedical/hmftools/blob/master/purple/DriverCatalog.md).

### Statistics and reproducibility
Statistical tests were performed as described in the figure legends using GraphPad Prism (v.9.5.0). Sample sizes, statistical analyses and significance values are reported in the figure legends, denoted in the figure panel or described in the text. $P \le 0.05$ was considered to be statistically significant. Error bars represent s.e.m. unless otherwise stated. Experiments showing representative images were independently repeated two (Fig. 4g and Extended Data Figs. 1a,e,g and 7a,b), three (Extended Data Fig. 5i) or four (Fig. 4j) times with similar results.

### Reporting summary
Further information on research design is available in the Nature Portfolio Reporting Summary linked to this article.

## Data availability
All RNA-seq data generated in this study have been deposited at the European Nucleotide Archive under accession number PRJEB59247. PCAWG analysis results, including somatic copy number and rearrangement calls, are available at https://dcc.icgc.org/releases/PCAWG. Raw, uncropped images of western blots are provided in the Supplementary Information. Source data are provided with this paper.

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

**Acknowledgements** We thank H. Skaletsky for advice on targeting the DYZ1 array; H. Yu, K.M. Dean and the UTSW-UNC Center for Cell Signaling Analysis (RM1GM145399) for assistance with microscopy; K. Jaqaman for advice on imaging analysis; J. W. C. Leung for providing reagents; and the members of the Ly Laboratory for discussions. We acknowledge the UT Southwestern Flow Cytometry Core for access to equipment and assistance. This work was supported by the US National Institutes of Health (R35GM146610 and R00CA218871 to P.L.; T32CA124334 to J.T.S.), the US Department of Defense (W81XWH2210764 to P.L.), the Cancer Prevention and Research Institute of Texas (RR180050 to P.L.; RP21004 to E.G.M.), the Welch Foundation (I-2071-20210327 to P.L.) and American Cancer Society Institutional Research Grant (ACS-IRG-21-142-16 to P.L.). I.C.-C. and J.E.V.-I. acknowledge the European Molecular Biology Laboratory for funding.

**Author contributions** Y.-F.L., Q.H. and P.L. conceived the project. Y.-F.L., Q.H., A.M. and P.L. designed the experiments. Y.-F.L., Q.H. and A.M. conducted most of the cell biological experiments and analysed the data. E.G.M. conducted ecDNA experiments. R.D. conducted RPTEC experiments. J.L.E. assisted with generating cell lines. A.G. and G.N. provided technical assistance. J.T.S. and I.C.-C. performed analyses of RNA-seq data. J.E.V.-I., I.C.-C. and P.L. conceived and designed methods to detect balanced chromothripsis using whole-genome sequencing data. D.B. and S.F.B. provided critical input. I.C.-C. and P.L. supervised the study. P.L. wrote the manuscript with input from all of the authors.

**Funding** Open access funding provided by European Molecular Biology Laboratory (EMBL).

**Competing interests** S.F.B. owns equity in, receives compensation from and serves as a consultant and on the scientific advisory board and board of directors of Volastra Therapeutics. The other authors declare no competing interests.

**Additional information**
**Correspondence and requests for materials** should be addressed to Isidro Cortés-Ciriano or Peter Ly.

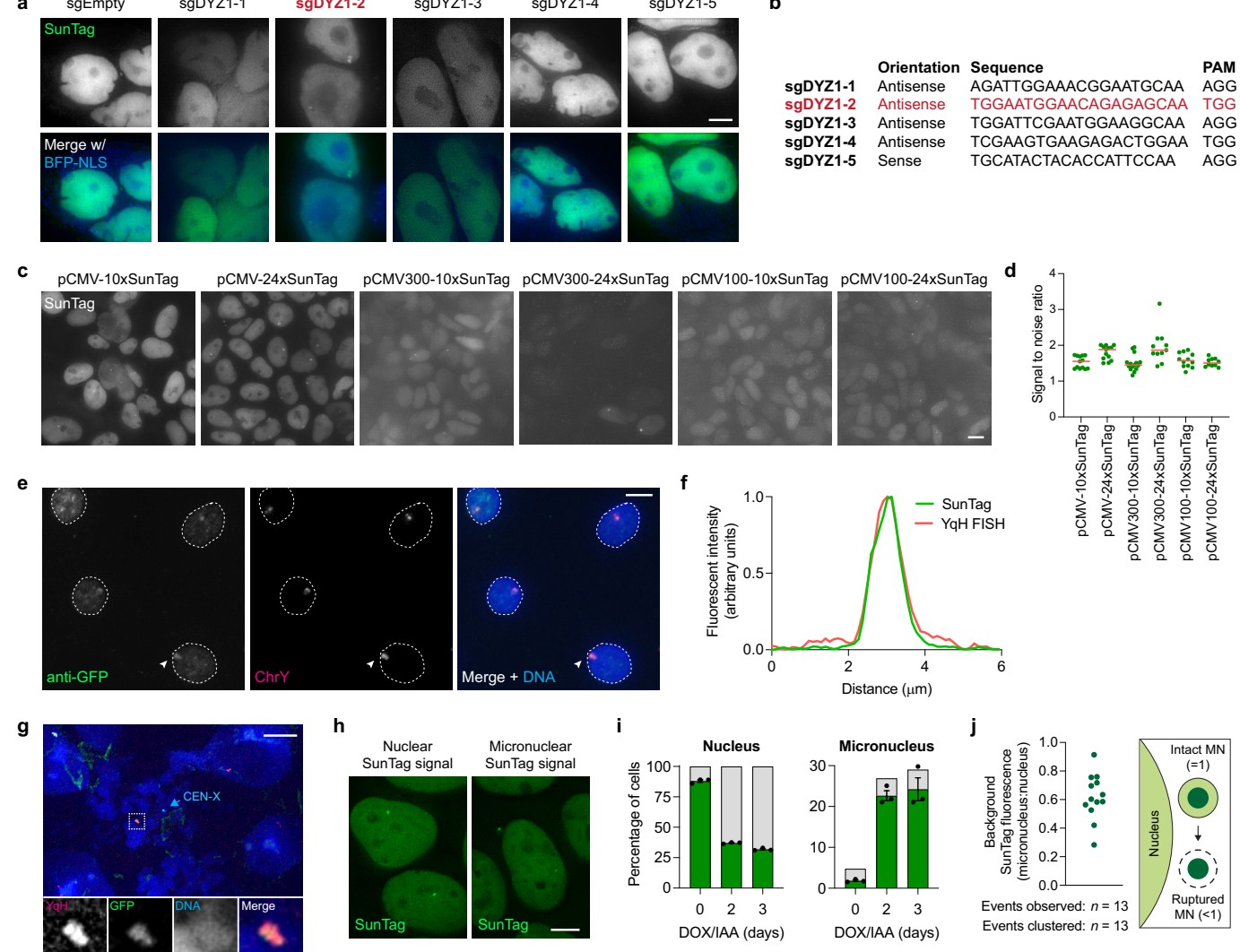

**b**

| | Orientation | Sequence | PAM |
|---|---|---|---|
| **sgDYZ1-1** | Antisense | AGATTGGAAACGGAATGCAA | AGG |
| **sgDYZ1-2** | Antisense | TGGAATGGAACAGAGAGCAA | TGG |
| **sgDYZ1-3** | Antisense | TGGATTCGAATGGAAGGCAA | AGG |
| **sgDYZ1-4** | Antisense | TCGAAGTGAAGAGACTGGAA | TGG |
| **sgDYZ1-5** | Sense | TGCATACTACACCATTCCAA | AGG |

**Extended Data Fig. 1 | Development of a live-cell Y chromosome-labelling system by targeting dCas9-SunTag to the DYZ1 array. a)** Images of DLD-1 cell populations expressing dCas9-SunTag and sfGFP-scFv with the indicated sgRNAs targeting the DYZ1 array. Scale bar, 5 μm. **b)** List of sgRNA sequences used in (**a**). sgDYZ1-2 was used for the remainder of the study. **c)** Images of DLD-1 cell populations expressing sfGFP-scFv under the control of full-length or truncated CMV promoters with dCas9-SunTag containing the indicated scaffold lengths. Scale bar, 5 μm. **d)** Signal-to-noise measurements for the conditions shown in (**c**). Data represent mean; from left to right, $n$ = 13, 13, 15, 11, 12, and 10 cells. **e)** IF-FISH image of interphase cells showing co-localization between an anti-GFP antibody recognizing sfGFP bound to dCas9-SunTag and DNA FISH probes targeting the Y chromosome q-arm heterochromatic array (YqH). Scale

bar, 5 μm. **f)** Fluorescent line scan analysis of the indicated region marked in (**e**) showing high specificity of the SunTag with YqH FISH. **g)** IF-FISH image of mitotic chromosomes showing co-localization between an anti-GFP antibody recognizing dCas9-SunTag with chromosome paint probes targeting YqH. Scale bar, 10 μm. **h)** Example images of live DLD-1 cells with dCas9-SunTag signals in the nucleus or micronucleus. Scale bar, 5 μm. **i)** Proportion of nuclei and micronuclei with or without dCas9-SunTag signals following DOX/IAA induction for the indicated number of days. Data represent mean ± SEM of $n$ = 3 independent experiments; 0 days = 1,044, 2 days = 1,070, 3 days = 1,123 cells. **j)** Background fluorescence measurements of non-dCas9-SunTag-bound sfGFP-scFv from $n$ = 13 micronuclei obtained from independent experiments (left) and schematic of intact and ruptured micronuclei measurements (right).

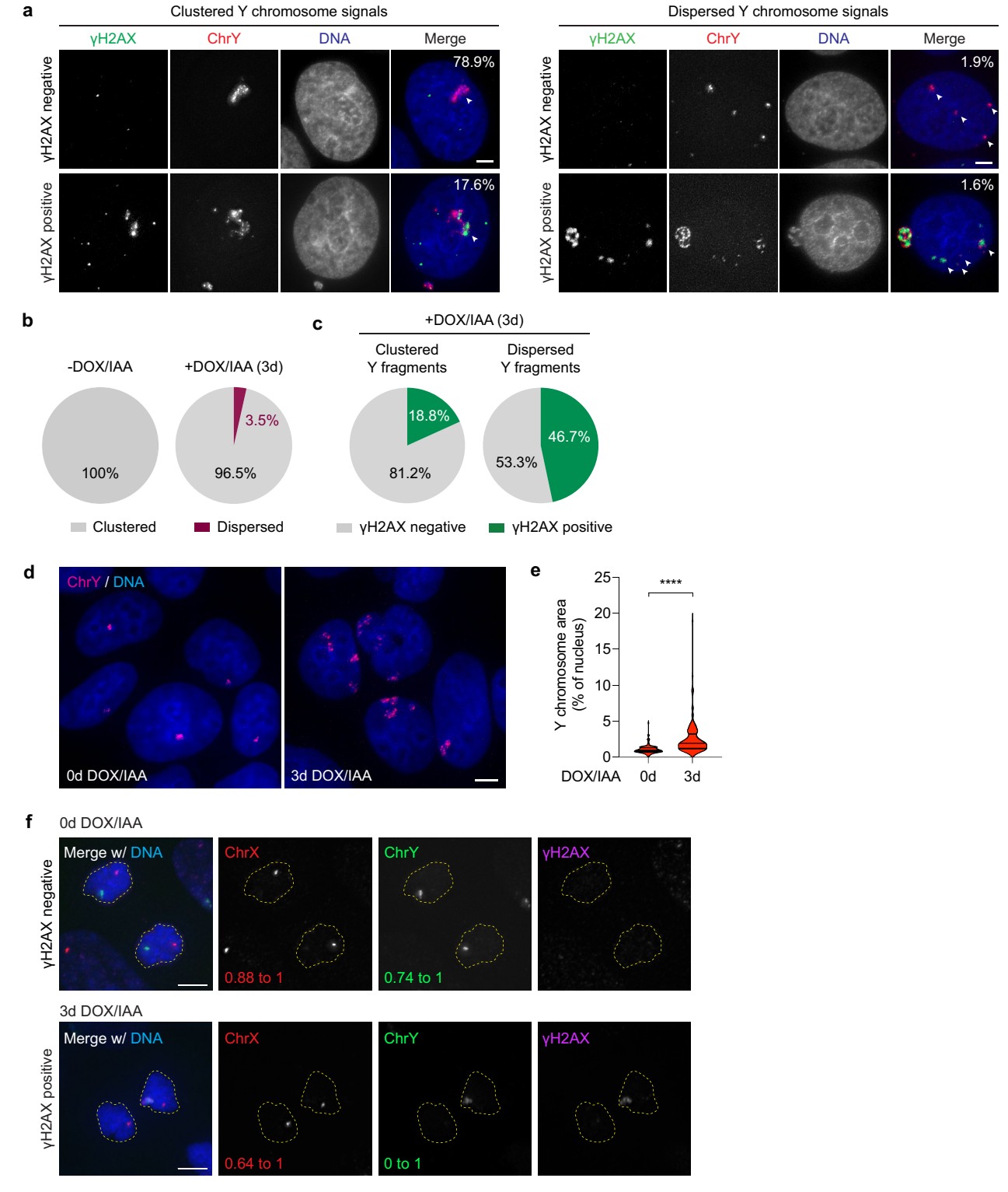

**Extended Data Fig. 2 | Micronucleation produces clusters of damaged chromosome fragments that unequally reincorporate into daughter cell nuclei. a)** Examples of clustered (left panels) and dispersed (right panels) Y chromosome signals in the interphase nucleus with or without γH2AX. Percentages shown represent the proportion of cells that exhibit each category following 3d DOX/IAA treatment. Scale bar, 5 µm. **b)** Pie charts depicting the fraction of control or DOX/IAA-treated cells with clustered or dispersed Y chromosome fragments during interphase. Data pooled from 2 independent experiments; -DOX/IAA: $n = 376$, +DOX/IAA: $n = 858$ cells. **c)** Pie charts depicting the γH2AX status of clustered and dispersed Y chromosome fragments. Data pooled from 2 independent experiments; clustered: $n = 828$, dispersed: $n = 30$ fragments. **d)** Re-integrated Y chromosome fragments occupy a larger nuclear space with increased nuclear fluorescence signal area following DOX/IAA treatment. Scale bar, 5 µm. **e)** Violin plot quantification of (**d**) measuring Y chromosome FISH area over the total area of the nucleus, as indicated by DAPI staining; data pooled from 0d: $n = 103$, 3d: $n = 105$ Y chromosome clusters; ****$P \leq 0.0001$ by Welch's two-tailed unpaired t-test. **f)** Images of equal segregation of an intact Y chromosome (top) and pulverized Y chromosome exhibiting unequal partitioning between daughter cells. Scale bar, 5 µm. Quantification shown in Fig. 1g.

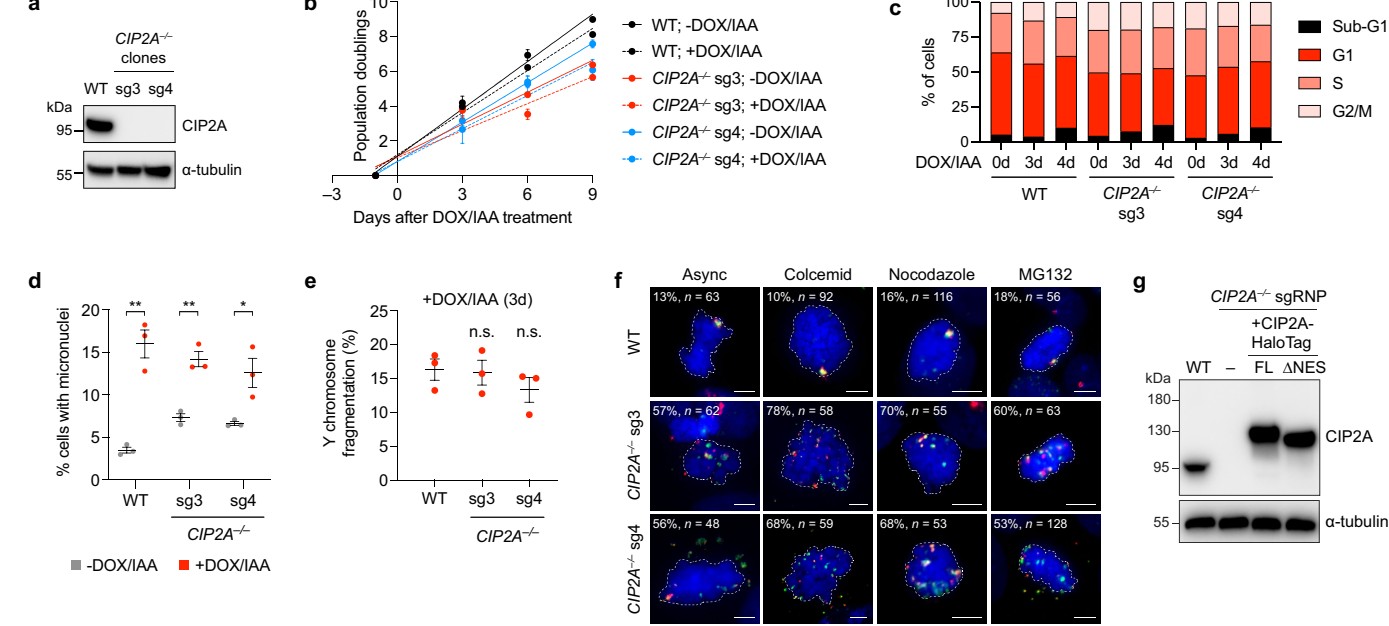

**Extended Data Fig. 3 | Characterization of human DLD-1 cells harbouring biallelic deletions in *CIP2A*. a)** Immunoblot confirmation of CIP2A KO clones. **b)** Growth curves of WT and CIP2A KO cells with and without DOX/IAA treatment over the indicated number of days. Data represent mean ± SEM; $n$ = 3 biological replicates. **c)** Flow cytometry analysis of propidium iodide-stained WT and CIP2A KO cells showing similar cell cycle distribution profiles. **d)** Proportion of micronucleated cells with or without 2d DOX/IAA induction, as determined by DAPI staining. Data represent mean ± SEM; WT: **$P$ = 0.0017, sg3: **$P$ = 0.0023, sg4: *$P$ = 0.0261 by two-tailed t-test compared to untreated controls; $n$ = 3 independent experiments; WT (-DOX/IAA = 5,521, +DOX/IAA = 3,718), CIP2A KO sg3 (-DOX/IAA = 3,436, +DOX/IAA = 2,450), CIP2A KO sg4 (-DOX/IAA =

3,999, +DOX/IAA = 2,930 cells). **e)** Frequency of Y chromosome fragmentation among Y chromosome-positive metaphase spreads following 3d DOX/IAA induction. Data represent mean ± SEM; not significant (ns), $P$ > 0.05 by two-tailed t-test compared to WT controls; $n$ = 3 independent experiments; WT = 234, CIP2A KO sg3 = 269, CIP2A KO sg4 = 284 cells. **f)** Mitotic CIP2A KO cells exhibiting fragment dispersion with and without cell cycle arrest with the indicated mitotic inhibitors. Dispersion frequencies and the number of cells analysed are shown. **g)** Immunoblot of ectopic CIP2A-HaloTag complementation in CIP2A KO cells generated by a frameshift deletion in exon 3 induced by Cas9 ribonucleoprotein (sgRNP) delivery. FL, full length; NES, nuclear export signal.

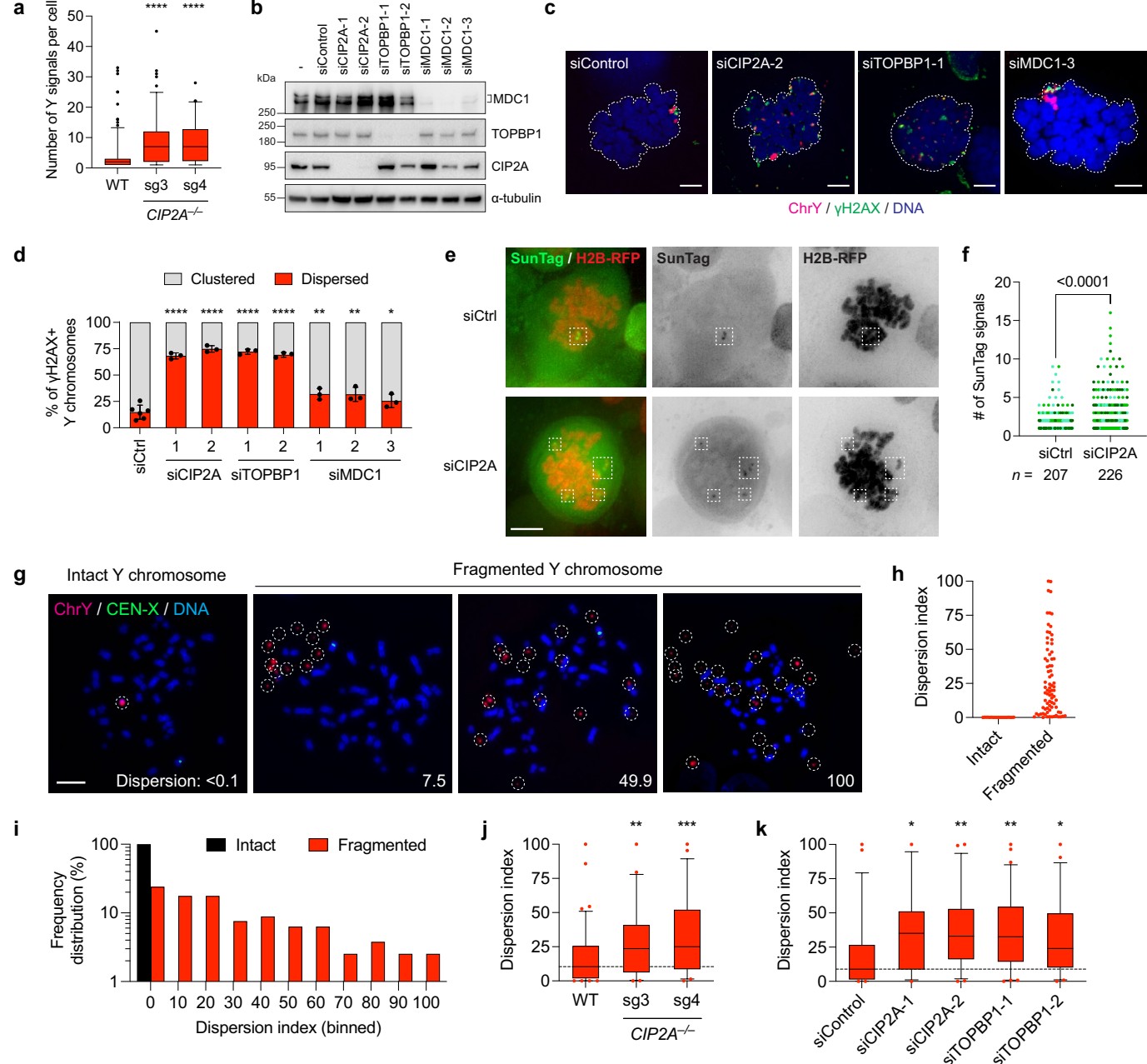

**Extended Data Fig. 4 | Loss of CIP2A-TOPBP1, but not MDC1, disperses fragmented chromosomes during mitosis. a)** Quantification of Y chromosome-positive signals in WT and CIP2A KO cells. Data represent median with 5-95 percentiles; ****$P \le 0.0001$ by two-tailed t-test compared to WT controls; WT: $n = 214$, CIP2A KO sg3: $n = 113$, CIP2A KO sg4: $n = 84$ cells pooled from 2 independent experiments. **b)** Immunoblot of CIP2A, TOPBP1, and MDC1 depletion in WT DLD-1 cells using two or three independent small interfering RNAs. Whole-cell extracts were collected 96 h after transfection. **c)** Images of fragmented Y chromosomes in mitotic WT DLD-1 cells depleted of CIP2A, TOPBP1, or MDC1 prior to DOX/IAA induction. Scale bar, 5 μm. **d)** Quantification of fragment clustering and dispersion from (**c**). Data represent the mean ± SEM of $n = 5$ (siCtrl) or 3 (all other conditions) independent experiments; *$P = 0.0495$, **$P = 0.001$, ****$P \le 0.0001$ by one-way ANOVA with multiple comparisons test compared to siControl sample; siControl = 450, siCIP2A-1 = 145, siCIP2A-2 = 178, siTOPBP1-1 = 138, siTOPBP1-2 = 133, siMDC1-1 = 298, siMDC1-2 = 386, siMDC1-3 = 281 cells. **e)** Live-cell images of dCas9-SunTag signals from nocodazole-arrested DLD-1 cells showing increased SunTag-positive fragments following CIP2A depletion. Scale bar, 5 μm. **f)** Quantification of the number of SunTag-positive signals from (**e**). Individual data points represent a single cell; data pooled from 3 independent experiments; two-tailed unpaired t-test with Welch's correction.

**g)** Metaphase spreads were collected from DLD-1 cells treated with DOX/IAA and hybridized to Y chromosome FISH probes. Examples of intact and fragmented Y chromosomes are shown along with dispersion index (see Methods for measurements). Scale bar, 10 μm. **h)** Quantification of metaphase fragment dispersion from (**g**). Data represent individual metaphase spreads with an intact or fragmented Y chromosome; intact: $n = 19$, fragmented: $n = 79$ metaphases from 3 independent experiments. **i)** Distribution of dispersion indices for intact and fragmented Y chromosomes from data shown in (**h**); intact: $n = 19$, fragmented: $n = 79$ metaphases from 3 independent experiments. **j)** CIP2A KO cells exhibit increased fragment dispersion on metaphase chromosome spreads. Data represent median with 5-95 percentiles; **$P = 0.0042$, ***$P = 0.0002$ by one-way ANOVA with multiple comparisons test compared to WT control sample; WT: $n = 83$, sg3: $n = 54$, sg4: $n = 44$ metaphases from 3 independent experiments. **k)** CIP2A or TOPBP1 depletion increases fragment dispersion on metaphase chromosome spreads. Data represent median with 5-95 percentiles; *$P \le 0.05$; **$P \le 0.01$ by one-way ANOVA with multiple comparisons test compared to siControl sample; siControl: $n = 57$, siCIP2A-1: $n = 38$, siCIP2A-2: $n = 52$, siTOPBP1-1: $n = 63$, siTOPBP1-2: $n = 49$ metaphases from 3 independent experiments.

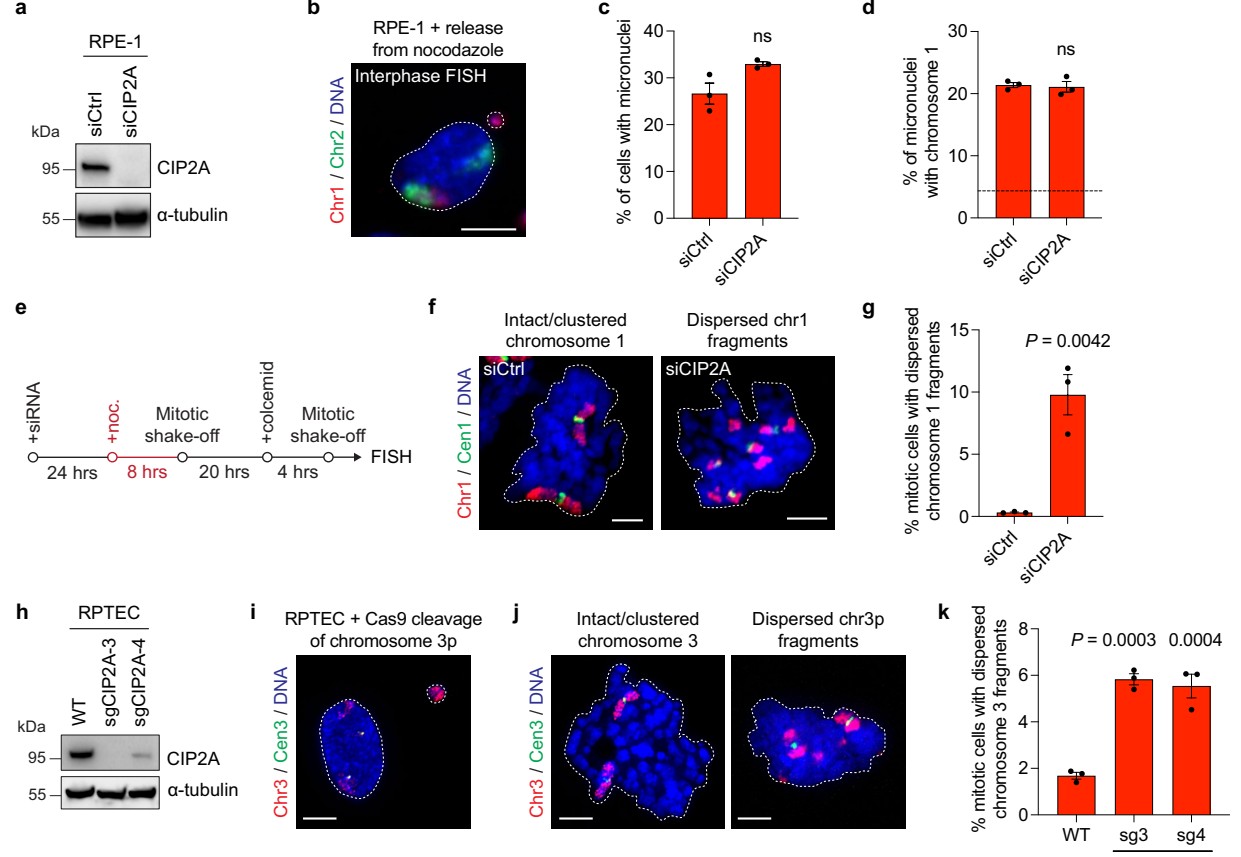

**Extended Data Fig. 5 | CIP2A-mediated mitotic clustering in additional human cell lines with distinct sources of micronuclei. a)** Immunoblot confirmation of RPE-1 cells depleted of CIP2A 72 h after transfection. **b)** Image of RPE-1 cell harbouring a micronucleus containing chromosome 1. RPE-1 cells were arrested in mitosis using nocodazole, released into interphase, and hybridized to the indicated chromosome paint probes by FISH. Scale bar, 10 μm. **c)** Quantification of micronuclei frequencies in nocodazole-arrested RPE-1 cells depleted of CIP2A. Data represent mean ± SEM; not significant (ns), $P > 0.05$ by two-tailed t-test compared to siCtrl; $n = 3$ independent experiments; siCtrl = 1,230, siCIP2A = 1,096 cells. **d)** Proportion of micronuclei containing chromosome 1. Data represent mean ± SEM; not significant (ns), $P > 0.05$ by two-tailed t-test compared to siCtrl; $n = 3$ independent experiments; siCtrl = 334, siCIP2A = 359 micronuclei. Dotted line represents frequency expected by random chance. **e)** Schematic to measure mitotic clustering of chromosome 1 fragments following CIP2A depletion and induction of chromosome 1 micronuclei in RPE-1 cells. **f)** Images of mitotic cells containing an intact

chromosome 1 or dispersed chromosome 1 fragments. Scale bar, 5 μm. **g)** Proportion of mitotic RPE-1 cells with visible chromosome 1 fragments following induction of chromosome 1 micronuclei. Data represent mean ± SEM; P-value derived from two-tailed t-test compared to siCtrl; $n = 3$ independent experiments; siCtrl = 997, siCIP2A = 1,361 mitotic cells. **h)** Immunoblot confirmation of RPTEC populations transduced with the indicated CRISPR lentiviruses. **i)** Image of RPTEC harbouring a micronucleus containing chromosome 3p. Cas9 ribonucleoproteins were delivered into RPTECs to induce a DSB on the chromosome 3p arm near the centromere in the presence of a DNA-PK inhibitor. Scale bar, 5 μm. **j)** Images of mitotic cells containing an intact chromosome 3p or dispersed chromosome 3p fragments. Scale bar, 5 μm. **k)** Proportion of mitotic RPTECs with visible chromosome 3p fragments following induction of chromosome 3p micronuclei. Data represent mean ± SEM; P-values derived from one-way ANOVA with multiple comparisons test compared to WT cells; $n = 3$ independent experiments; WT = 1,050, sg3 = 1,223, sg4 = 1,236 mitotic cells.

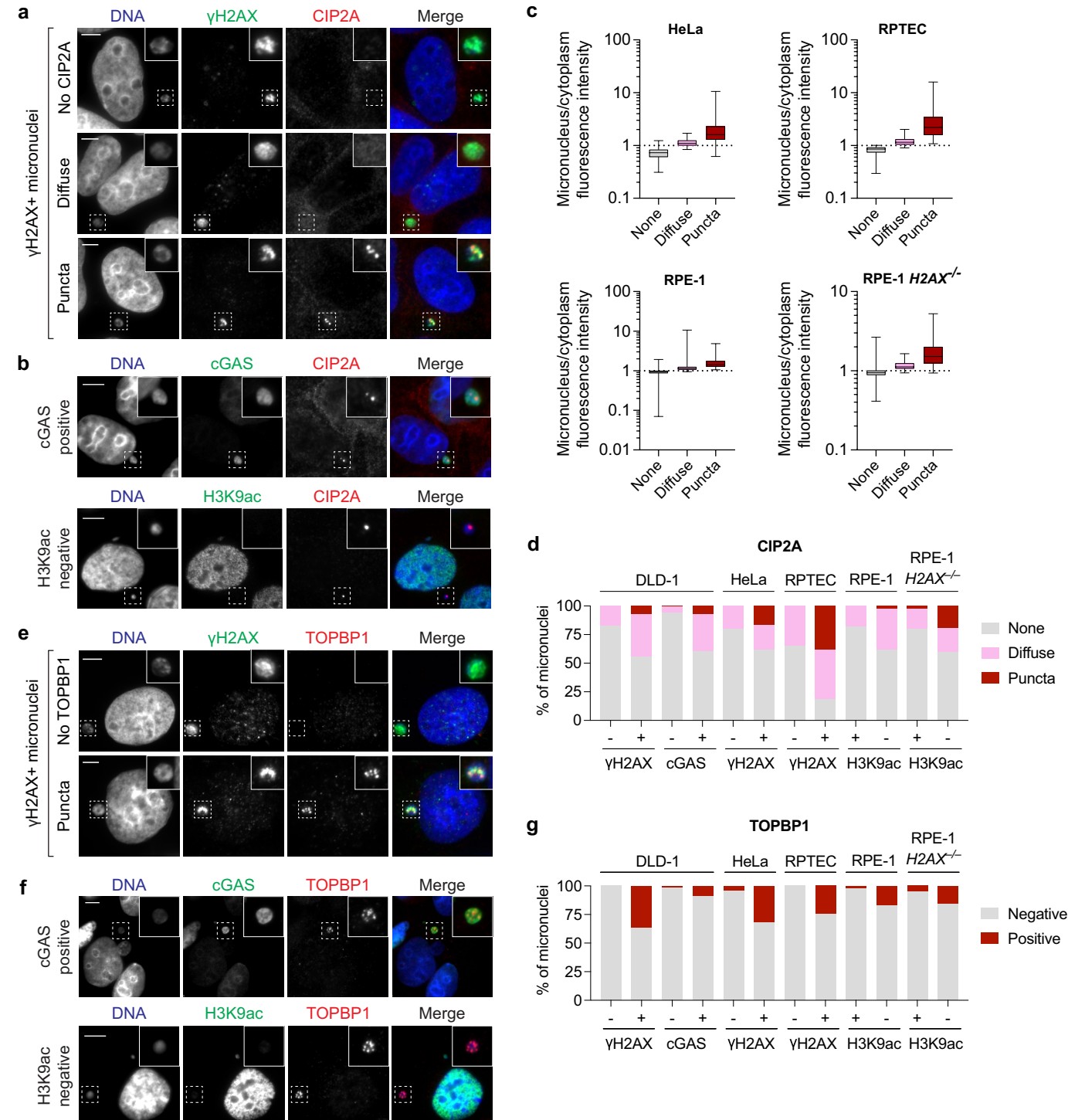

**Extended Data Fig. 6** | See next page for caption.

**Extended Data Fig. 6 | Premature interphase recruitment of CIP2A-TOPBP1 to ruptured micronuclei. a)** Examples of CIP2A localization patterns in ruptured (γH2AX-positive) micronuclei of DLD-1 cells. Intensity measurements shown in Fig. 3a. Scale bar, 10 μm. **b)** Examples of CIP2A localization patterns in ruptured micronuclei in DLD-1 cells expressing cGAS-GFP and immunostained for cGAS accumulation (top) or the lack of acetylated H3K9 (bottom) in RPE-1 cells. Scale bar, 10 μm. **c)** Intensity measurements of distinct CIP2A localization patterns in micronuclei compared to the cytoplasm in the indicated cell lines. Box plot represents interquartile range with min-max; HeLa: none, $n = 322$, diffuse, $n = 48$, puncta, $n = 30$; RPTEC: none, $n = 91$, diffuse, $n = 100$, puncta, $n = 69$; RPE-1: none, $n = 317$, diffuse, $n = 86$, puncta, $n = 34$; RPE-1 $H2AX^{-/-}$: none, $n = 274$, diffuse, $n = 47$, puncta, $n = 40$ micronuclei pooled from 3-5 independent experiments. **d)** Frequency of CIP2A localization patterns in ruptured (γH2AX-positive, cGAS-positive, or H3K9ac-negative) micronuclei across a panel of human cell lines. Data represent mean; from left to right, $n = 218, 99, 169, 134,$ 119, 76, 111, 211, 196, 92, 121, and 67 micronuclei pooled from 2 (RPE-1 $H2AX^{-/-}$) or 3 (all other conditions) independent experiments. **e)** Examples of TOPBP1 localization patterns in ruptured (γH2AX-positive) micronuclei of DLD-1 cells. Scale bar, 10 μm. **f)** Examples of TOPBP1 localization patterns in ruptured micronuclei in DLD-1 cells expressing cGAS-GFP and immunostained for cGAS accumulation (top) or the lack of acetylated H3K9 (bottom) in RPE-1 cells. Scale bar, 10 μm. **g)** Frequency of TOPBP1 localization patterns in ruptured (γH2AX-positive, cGAS-positive, or H3K9ac-negative) micronuclei across a panel of human cell lines. Data represent mean; from left to right, $n = 323, 120, 172, 133,$ 143, 87, 160, 295, 221, 82, 230, and 88 micronuclei pooled from 3 independent experiments. For (**d**) and (**g**), DLD-1 cells were treated with DOX/IAA to induce Y chromosome micronuclei, HeLa and RPE-1 cells were treated with CENP-E/ Mps1 inhibitors to induce random micronuclei, and RPTECs were transfected with Cas9 ribonucleoproteins targeting the chromosome 3p arm near the centromere to induce chromosome 3p micronuclei.

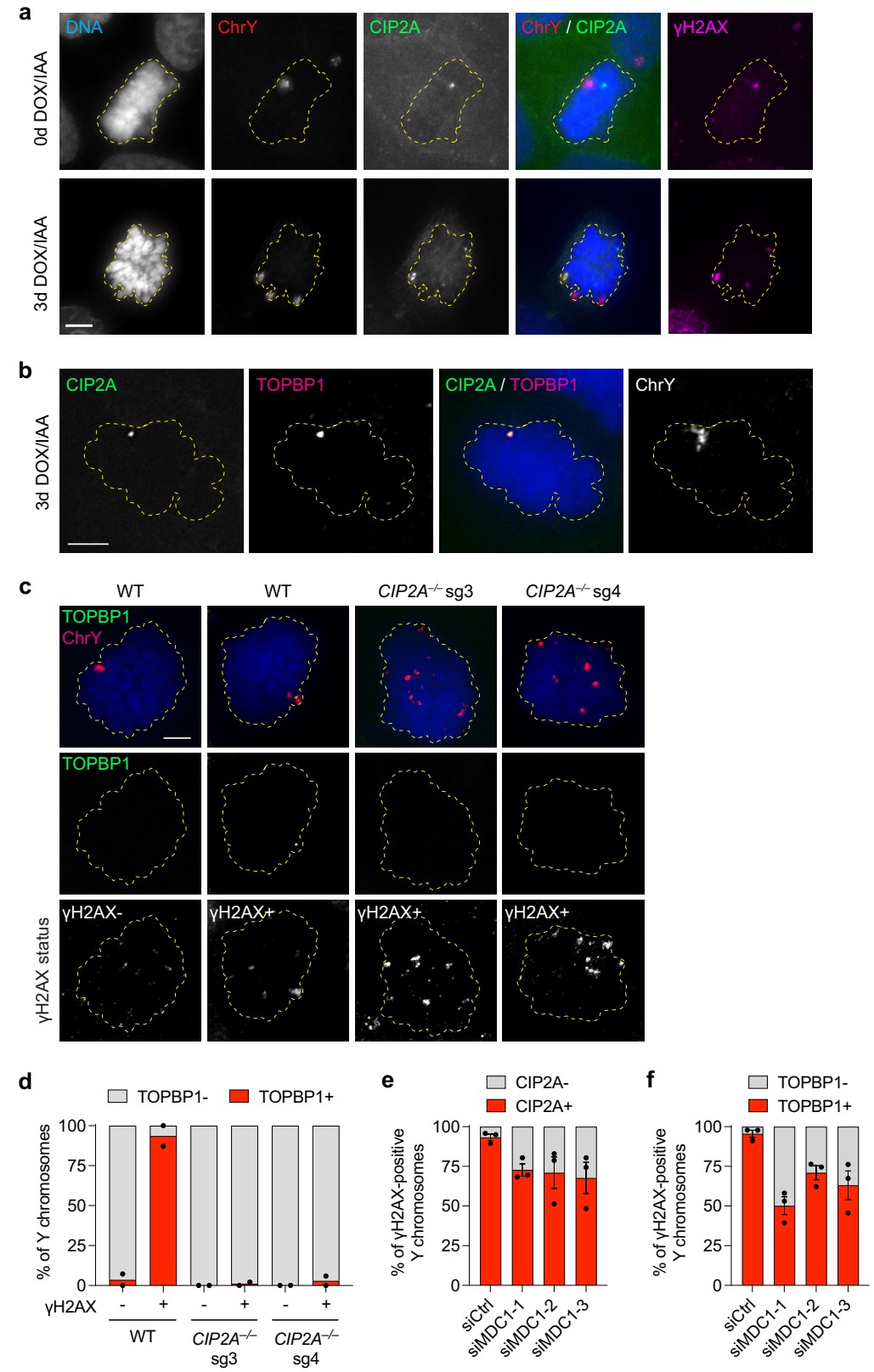

**Extended Data Fig. 7** | See next page for caption.

**Extended Data Fig. 7 | Mitotic localization of CIP2A-TOPBP1 on pulverized chromosomes. a)** Mitotic DLD-1 cells stained for CIP2A and H2AX and hybridized to chromosome paint probes. In untreated cells, CIP2A specifically co-localizes with spontaneous DNA lesions. Scale bar, 5 μm. **b)** Mitotic DLD-1 cell stained for CIP2A and TOPBP1 and hybridized to chromosome paint probes showing co-localization between CIP2A-TOPBP1 with the Y chromosome. Scale bar, 5 μm. **c)** Mitotic WT or CIP2A KO DLD-1 cells stained for TOPBP1 and H2AX and hybridized to chromosome paint probes. CIP2A loss prevents TOPBP1 recruitment to dispersed Y chromosome fragments. Scale bar, 5 μm. **d)** Quantification of (**c**). Data represent the mean of $n = 2$ independent experiments; left to right: 330, 114, 94, 124, 308, and 153 cells. **e)** Quantification of CIP2A localization to clustered mitotic chromosome fragments following MDC1 depletion. Data represent the mean ± SEM of $n = 3$ independent experiments; left to right: 153, 123, 162, and 116 cells. **f)** Quantification of TOPBP1 localization to clustered mitotic chromosome fragments following MDC1 depletion. Data represent the mean ± SEM of $n = 3$ independent experiments; left to right: 136, 125, 155, and 137 cells.

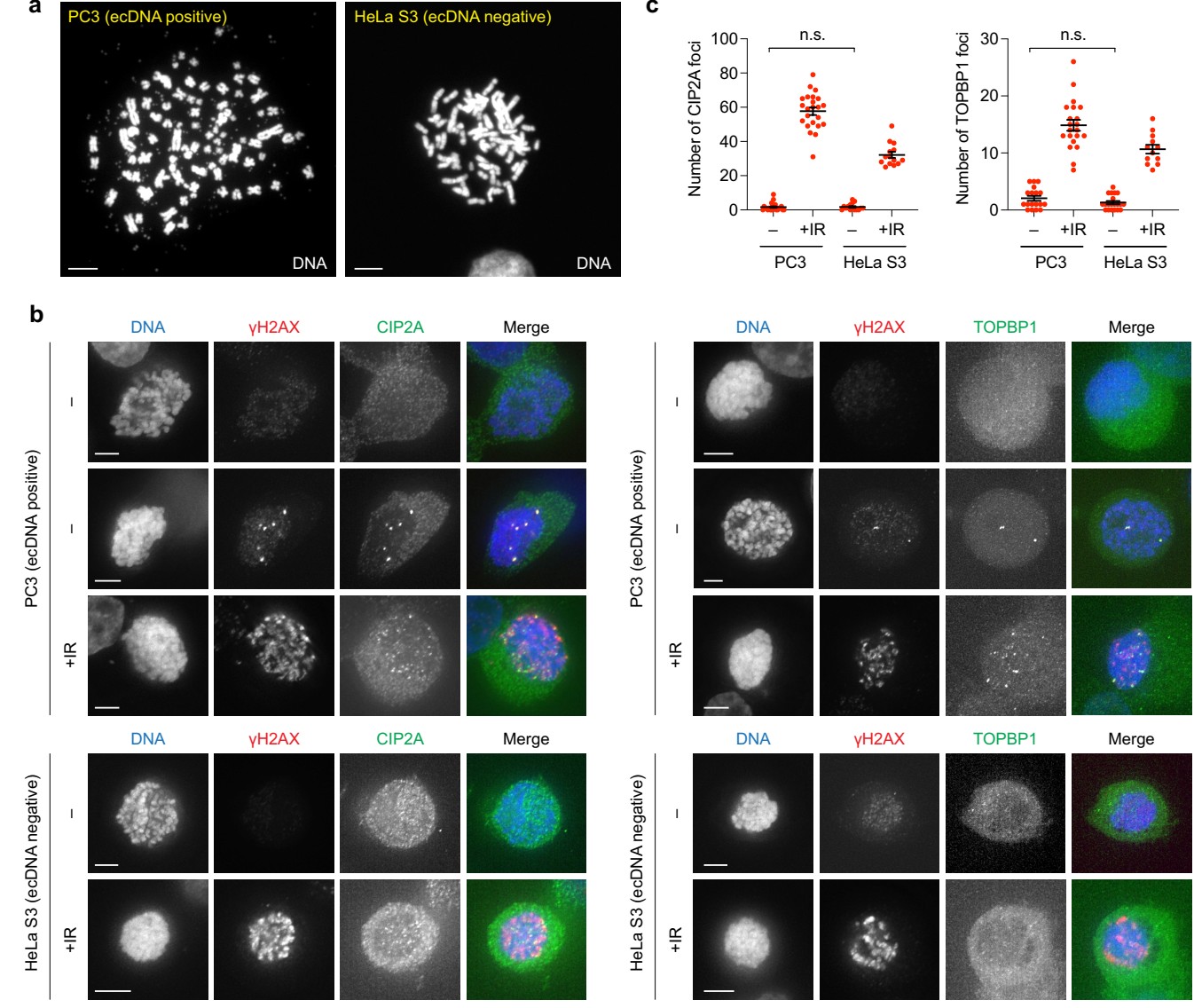

**Extended Data Fig. 8 | CIP2A-TOPBP1 does not associate with acentric extrachromosomal DNA (ecDNA) elements. a)** DAPI-stained metaphase spreads showing abundant ecDNAs in PC3 cells but not control HeLa S3 cells. Scale bar, 5 µm. **b)** CIP2A and TOPBP1 are not recruited to mitotic chromosomes in PC3 cells with ecDNAs in the absence of DNA damage. Examples of untreated and irradiated PC3 and HeLa cells arrested in mitosis and immunostained for CIP2A or TOPBP1. Scale bar, 5 µm. **c)** Quantification of CIP2A and TOPBP1 foci in (**b**). Data represent the mean ± SEM; $P = 0.8768$ (ns) for CIP2A; from left to right, $n = 24, 23, 15,$ and 14 mitotic cells; $P = 0.1437$ (ns) for TOPBP1; from left to right, $n = 18, 21, 20,$ and 12 mitotic cells; $P$-values calculated by two-tailed t-test comparing non-irradiated PC3 and HeLa S3 cells.

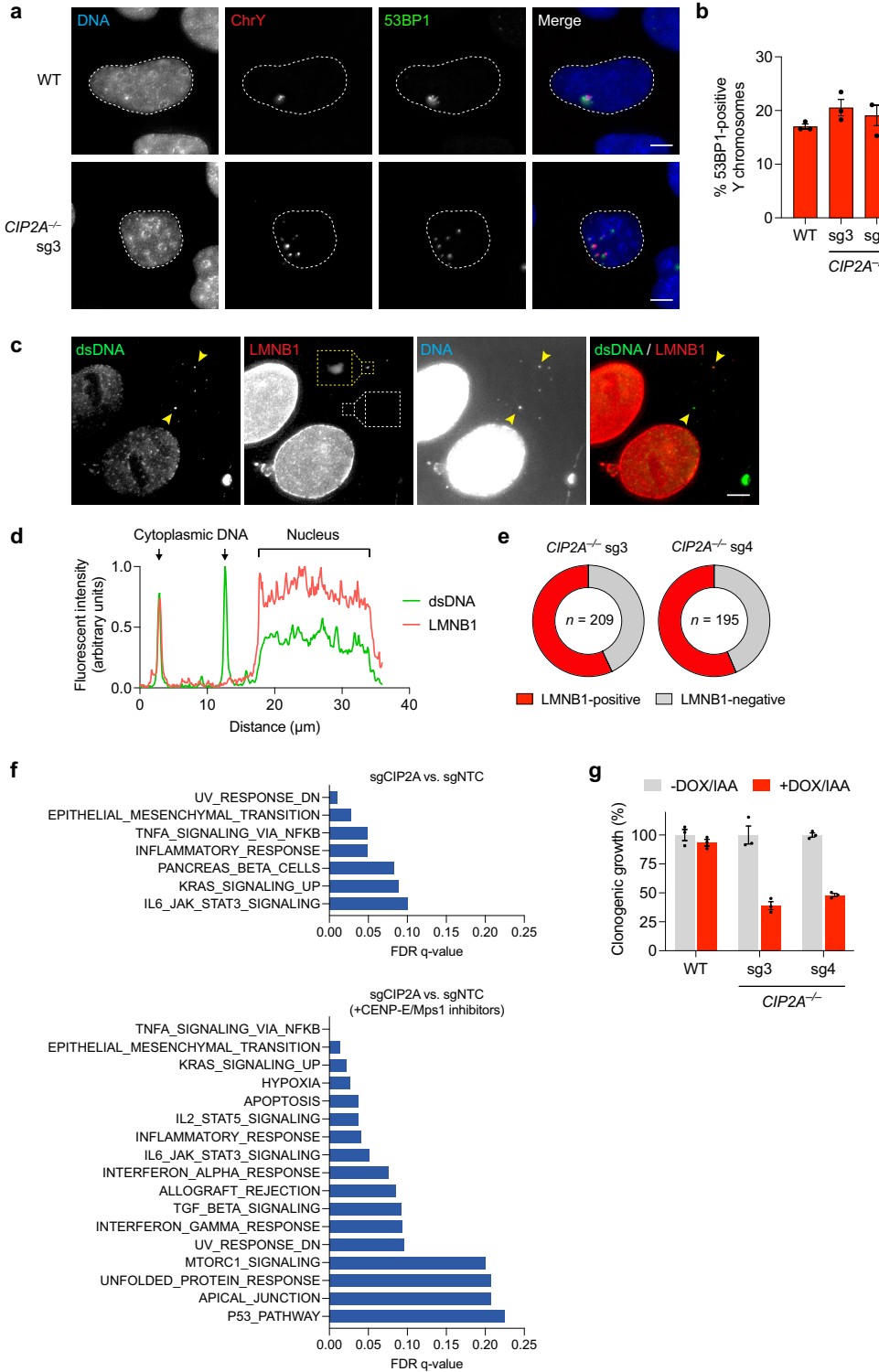

**Extended Data Fig. 9 | Dispersed nuclear and cytoplasmic DNA fragments activate DNA damage signalling and inflammatory responses, respectively.** **a)** 53BP1 immunostaining reveals engagement of clustered (top) and dispersed (bottom) nuclear fragments by the DNA damage response. **b)** Frequency of cells with 53BP1-positive Y chromosomes, as determined by IF-FISH. Data represent mean ± SEM of $n$ = 3 independent experiments; WT: 770, sg3: 735, sg4: 502 cells. **c)** Examples of cytoplasmic DNA foci that are positive (yellow box, see magnified inset) or negative (white box) for the nuclear membrane marker lamin B1. Scale bar, 5 μm. **d)** Fluorescent intensity line scan analysis between the indicated arrows depicted in (**c**) showing examples of cytoplasmic DNA foci with and without lamin B1. **e)** Proportion of cytoplasmic DNA foci with

and without lamin B1 staining from (**c**). Pie charts represent mean; $n$ = number of foci pooled from 2 independent experiments. **f)** Gene set enrichment analysis of bulk RNA sequencing of two HeLa cell populations individually transduced with two CIP2A sgRNAs (sgCIP2A) versus a non-targeting control sgRNA (sgNTC) with and without the induction of micronuclei using CENP-E/Mps1 inhibitors. Hallmark pathways with false-discovery rate (FDR) q-values < 0.25 are shown in ranked order. RNA sequencing was performed on three independent replicates per condition. **g)** Single-cell clonogenic growth assays showing that CIP2A KO cells, but not WT cells, are sensitive to the induction of micronuclei. Data represent mean ± SEM of $n$ = 3 biological replicates.

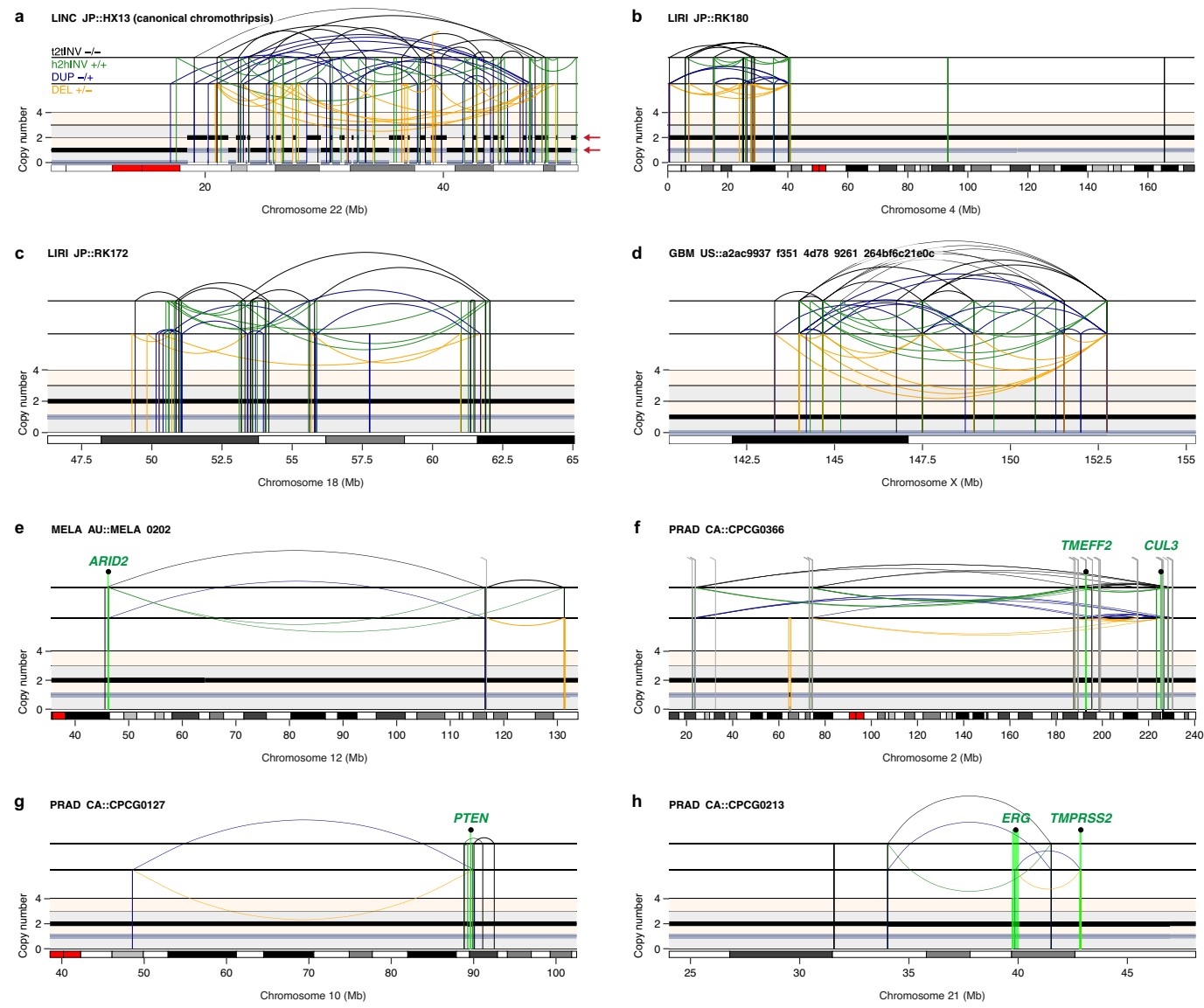

**Extended Data Fig. 10 | Examples of balanced chromothripsis events in cancer genomes. a)** Example of a canonical chromothripsis event affecting the q-arm of chromosome 22 in liver hepatocellular carcinoma exhibiting the characteristic pattern of DNA copy number oscillations (indicated by the red arrows). **b–f)** Additional examples of balanced chromothripsis events detected in liver cancer (**b–c**), glioblastoma (**d**), melanoma (**e**), and prostate adenocarcinoma (**f**). **g–h)** Examples of balanced chromothripsis events causing inactivation of *PTEN* (**g**) and generating a *TMPRSS2-ERG* gene fusion (**h**) in prostate adenocarcinomas.

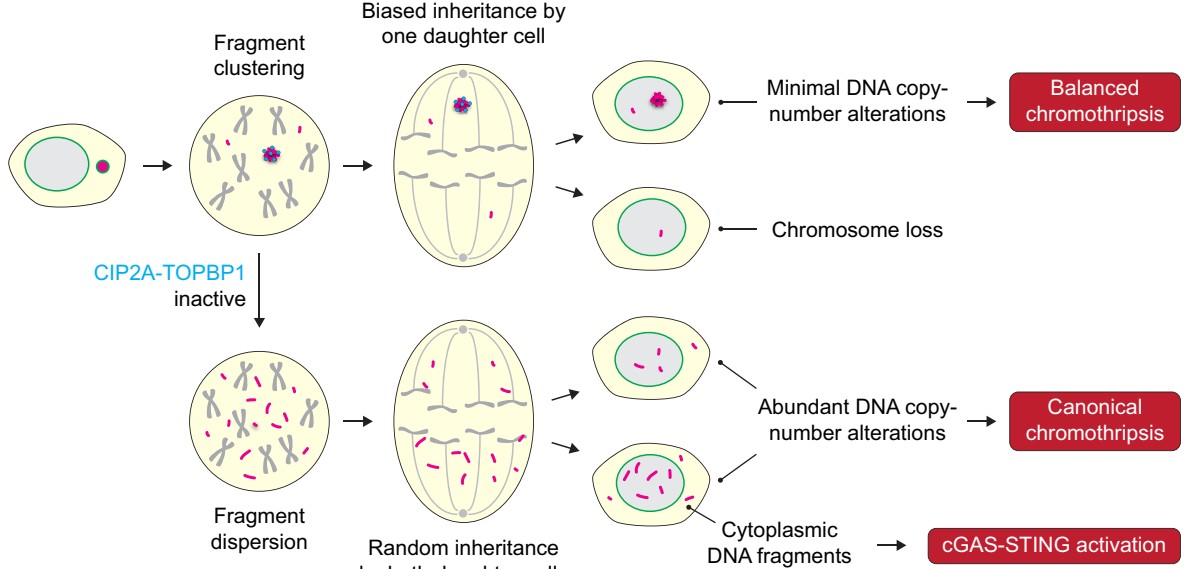

**Extended Data Fig. 11 | Mitotic clustering underlies distinct patterns of rearrangements following chromothripsis in micronuclei.** CIP2A-TOPBP1-mediated mitotic clustering of pulverized chromosomes from micronuclei facilitates balanced rearrangements in one of the daughter cells following mother cell division. In the absence of CIP2A-TOPBP1, pulverized fragments disperse throughout the mitotic cytoplasm and stochastically partition into both daughter cells.

# Reporting Summary

## Statistics

For all statistical analyses, confirm that the following items are present in the figure legend, table legend, main text, or Methods section.

| n/a | Confirmed | |
|---|---|---|
| ☐ | ☒ | The exact sample size (*n*) for each experimental group/condition, given as a discrete number and unit of measurement |
| ☐ | ☒ | A statement on whether measurements were taken from distinct samples or whether the same sample was measured repeatedly |
| ☐ | ☒ | The statistical test(s) used AND whether they are one- or two-sided *Only common tests should be described solely by name; describe more complex techniques in the Methods section.* |
| ☒ | ☐ | A description of all covariates tested |
| ☐ | ☒ | A description of any assumptions or corrections, such as tests of normality and adjustment for multiple comparisons |
| ☐ | ☒ | A full description of the statistical parameters including central tendency (e.g. means) or other basic estimates (e.g. regression coefficient) AND variation (e.g. standard deviation) or associated estimates of uncertainty (e.g. confidence intervals) |
| ☐ | ☒ | For null hypothesis testing, the test statistic (e.g. *F*, *t*, *r*) with confidence intervals, effect sizes, degrees of freedom and *P* value noted *Give P values as exact values whenever suitable.* |
| ☒ | ☐ | For Bayesian analysis, information on the choice of priors and Markov chain Monte Carlo settings |
| ☒ | ☐ | For hierarchical and complex designs, identification of the appropriate level for tests and full reporting of outcomes |
| ☒ | ☐ | Estimates of effect sizes (e.g. Cohen's *d*, Pearson's *r*), indicating how they were calculated |

*Our web collection on statistics for biologists contains articles on many of the points above.*

## Software and code

Policy information about availability of computer code

| Data collection | All microscopy images were collected on a DeltaVision Ultra microscope system (GE Healthcare), a Metafer Scanning and Imaging Platform (MetaSystems), or an ImageXpress Confocal HT.ai High-Content Imaging System (Molecular Devices). RNA sequencing was performed on an Illumina NovaSeq 6000 platform by Novogene. <br><br> Software versions: Metafer 4, version 3.13.6, MetaSystems; softWoRx, version 7.2.1, Cytiva |
|---|---|
| Data analysis | Statistical analyses for cell biological experiments were performed with GraphPad Prism (version 9.5.0) as described in the figure legends. Statistical analyses for RNA sequencing was performed as described in the Methods. <br><br> Software versions: Fiji, version 2.1.0/1.53c; FlowJo, version 10.8.2, BD Biosciences; STAR, version 2.7.4a; HTSeq, version 0.6.1p1; GSEA, version 4.3.2 |

For manuscripts utilizing custom algorithms or software that are central to the research but not yet described in published literature, software must be made available to editors and reviewers. We strongly encourage code deposition in a community repository (e.g. GitHub). See the Nature Portfolio guidelines for submitting code & software for further information.

## Data

Policy information about availability of data

All manuscripts must include a data availability statement. This statement should provide the following information, where applicable:
- Accession codes, unique identifiers, or web links for publicly available datasets
- A description of any restrictions on data availability
- For clinical datasets or third party data, please ensure that the statement adheres to our policy

RNA sequencing data generated by this study are deposited with the European Nucleotide Archive under accession PRJEB59247.

## Human research participants

Policy information about studies involving human research participants and Sex and Gender in Research.

| | |
|---|---|
| Reporting on sex and gender | N/A |
| Population characteristics | N/A |
| Recruitment | N/A |
| Ethics oversight | N/A |

Note that full information on the approval of the study protocol must also be provided in the manuscript.

# Field-specific reporting

Please select the one below that is the best fit for your research. If you are not sure, read the appropriate sections before making your selection.

☒ Life sciences      ☐ Behavioural & social sciences      ☐ Ecological, evolutionary & environmental sciences

For a reference copy of the document with all sections, see nature.com/documents/nr-reporting-summary-flat.pdf

# Life sciences study design

All studies must disclose on these points even when the disclosure is negative.

| | |
|---|---|
| Sample size | Sample sizes were not predetermined but were chosen based on current practices in the field. Exact sample sizes and/or the number of independent experiments performed per experiment are indicated in the figure, figure legends, or methods. All experiments reporting P-values were performed independently at least three times. |
| Data exclusions | No data were excluded from analyses. |
| Replication | All experiments were independently conducted and reproduced multiple times, as described in the figure legends. All P-values were derived from measurements obtained from experiments conducted independently at least three times. Figures with representative images were reproduced and obtained from at least two or more independent experiments with similar results. |
| Randomization | N/A |
| Blinding | Investigators were not blinded during data collection and/or analysis as each series of experiments were performed by an individual researcher. |

# Reporting for specific materials, systems and methods

We require information from authors about some types of materials, experimental systems and methods used in many studies. Here, indicate whether each material, system or method listed is relevant to your study. If you are not sure if a list item applies to your research, read the appropriate section before selecting a response.

## Materials & experimental systems

| n/a | Involved in the study |
|-----|----------------------|
| ☐ | ☒ Antibodies |
| ☐ | ☒ Eukaryotic cell lines |
| ☒ | ☐ Palaeontology and archaeology |
| ☒ | ☐ Animals and other organisms |
| ☒ | ☐ Clinical data |
| ☒ | ☐ Dual use research of concern |

## Methods

| n/a | Involved in the study |
|-----|----------------------|
| ☒ | ☐ ChIP-seq |
| ☒ | ☐ Flow cytometry |
| ☒ | ☐ MRI-based neuroimaging |

## Antibodies

| | |
|---|---|
| Antibodies used | For immunofluorescence: 1:500 anti-CIP2A (sc-80659, Santa Cruz), 1:1,000 anti-CIP2A (14805, Cell Signaling), 1:500 anti-TOPBP1 (sc-271043, Santa Cruz), 1:300-500 anti-TOPBP1 (ABE1463, Millipore), 1:1,000 anti-phospho H2AX (S139) (05-636, Millipore), 1:1,000 anti-phospho H2AX (S139) (2577, Cell Signaling), 1:1,000 anti-53BP1 (NB100-304, Novus), 1:1,000 anti-acetyl-histone H3 (Lys 9) (9649, Cell Signaling), 1:1,000 anti-cGAS (15102, Cell Signaling). For immunoblotting: 1:1,000 anti-CIP2A (sc-80659, Santa Cruz), 1:5,000 anti-α-tubulin (3873, Cell Signaling), 1:1,000 anti-TOPBP1 (sc-271043, Santa Cruz), 1:1,000 anti-phospho-histone H3 Ser10 (06-570, Millipore), 1:5,000 anti-MDC1 (ab11171, Abcam). For secondary antibodies, Alexa Fluor-conjugated donkey anti-rabbit and donkey anti-mouse antibodies (Invitrogen) were used for immunofluorescence experiments; horseradish peroxidase-conjugated goat anti-rabbit and donkey anti-mouse antibodies (Invitrogen) were used for immunoblotting. |
| Validation | The primary antibodies used in this study are commercially available. All critical antibodies were validated by depletion or knockout of the target gene using RNA interference or CRISPR/Cas9 editing, respectively. |

## Eukaryotic cell lines

Policy information about cell lines and Sex and Gender in Research

| | |
|---|---|
| Cell line source(s) | DLD-1, human colorectal cancer cells; RPE-1, human retinal pigment epithelial cells; RPTEC, human renal proximal tubule epithelial cells; HeLa, human cervical cancer cells; PC3, human prostate cancer cells; 293T, human embryonic kidney cells; 293GP, human embryonic kidney cells. DLD-1, HeLa, 293T, and 293GP cells were obtained from the cell line repository of Don Cleveland, RPE-1 cells originally generated by Stephen Jackson were obtained through Justin Leung, RPTECs were obtained from Denise Marciano, and PC3 cells were obtained from Sihan Wu. |
| Authentication | Cell lines were authenticated by morphological characteristics, SNP array analysis, karyotyping, and/or whole-genome DNA sequencing when possible. |
| Mycoplasma contamination | All cell lines used in this study were routinely confirmed to be free of mycoplasma contamination using the Universal Mycoplasma Detection Kit (ATCC) and by routine DAPI staining. |
| Commonly misidentified lines (See ICLAC register) | No commonly misidentified cell lines were used in this study |

nature portfolio | reporting summary

March 2021

