## [Peer Review File · Nature]

Manuscript Title: Mitotic clustering of pulverized chromosomes from micronuclei

Reviewer Comments & Author Rebuttals

Reviewer Reports on the Initial Version:

Referees' comments:

Referee #1 (Remarks to the Author):

The manuscript by Lin and colleagues aims to uncover the mechanism that promotes the clustering of micronuclear chromatin fragments that are produced following rupture of micronuclei. They identify the newly described CIP2A-TOPBP1 complex as a mediator of micronuclear chromosome clustering. This mechanism promotes the biased inheritance of the micronuclear genome into a single daughter cell, which preserves copy number and is consistent with observation of balanced chromothripsis, which was first observed in germline chromothripsis. The authors finally provide evidence that balanced chromothripsis is also observed in tumors, and find that the chromosome rearrangements often target oncogenes and tumor suppressors.

Overall, this nicely written manuscript will have a clear impact on our understanding on the formation of micronuclei-driven genome rearrangements in genetic diseases and cancer. Furthermore, the work supports the idea that CIP2A-TOPBP1 acts to tether acentric chromosome fragments, and this cements the role of CIP2A-TOPBP1 in this exciting new biology. Finally, as the CIP2A-TOPBP1 complex was recently shown to be essential in BRCA1/BRCA2-deficient cells, this work provides another reason to consider this complex as a drug target as the loss of micronuclear chromosome clustering is proposed to boost cytosolic DNA sensing, which is a likely anti-tumor mechanism. The data is generally well controlled, uses very elegant approaches and the conclusions are well supported. I am enthusiastic about this work.

I only have a few points that hopefully will improve this already compelling manuscript.

MAIN POINTS

1) The localization of CIP2A and TOPBP1 in ruptured interphase micronuclei raises the possibility that their main function is not related to clustering in mitosis but rather a role in interphase micronuclei (such as DNA repair). Their studies with clonal KOs do not allow the authors to fully exclude this possibility (even if Fig 3e supports their model). Therefore, their conclusions would be strengthened if they could achieve mitotic-specific depletion of CIP2A, TOPBP1 or the CIP2A-TOPBP1 interaction and observe dispersion of micronuclear chromosome fragments in mitotic cells.

2) The CIP2A-TOPBP1 complex localizes to mitotic DSBs in an MDC1-dependent manner whereas it localizes to mitotic DNA lesions that arise from under replicated DNA independently of MDC1. The authors should test if the micronuclear CIP2A/TOPBP1 foci are MDC1-dependent, suggesting it recognizes DSBs formed in micronucle, or whether they form independently of MDC1, perhaps

pointing to defective DNA replication as the lesion(s) sensed by CIP2A-TOPBP1.

3) The authors show convincingly that micronuclear fragment clustering by CIP2A-TOPBP1 prevents accumulation of cytoplasmic dsDNA. It would be useful to test whether loss of CIP2A, TOPBP1 or the CIP2A-TOPBP1 interaction is also accompanied by stronger or prolonged innate immune response to micronuclei. This may strengthen the case that targeting the CIP2A-TOPBP1 interaction could be therapeutically attractive as an anti-cancer strategy.

4) The authors imply that CIP2A-TOPBP1 promotes the formation of balanced chromothripsis over the canonical kind. Testing this prediction experimentally would go a long way towards linking the mechanistic and cancer genomics aspects of the paper.

Referee #2 (Remarks to the Author):

The authors present an analysis of the role of the CIP2A-TOPBP1 complex in maintaining chromosomal fragments in close proximity to one another. One prediction of the model is that there should be examples of chromothripsis in which all chromosomal fragments are retained in the eventual reconstructed chromosome, and indeed the authors find examples of so-called 'balanced chromothripsis' in which there are no copy number changes despite the presence of multiple SVs suggestive of chromothripsis.

I enjoyed reading this manuscript, and there is much to like in the dissection of the role of CIP2A-TOPBP1 - I think this story is of interest to the general readership of Nature. I have two major comments that would strengthen the story -

1. The most important limitation of the story as currently presented, in my opinion, is the lack of a link between the human data and the CIP2A-TOPBP1 complex story - that is, the existence of balanced chromothripsis in human cancers does not substantiate the relevance of that complex to chromothripsis *in vivo*. More convincing evidence could come in the form of either (1) an analysis that showed that unbalanced chromothripsis was more common in cancers that had LOH / mutation of the complex or (2) an *in vitro* study of the DLD-1 cell line showing that the chromothripsis that emerged after CIP2A/TOPBP1 knockdown was more likely to be unbalanced than when the complex is intact.

2. The *in vitro* analysis is restricted to a single cell line with one mechanism of micronucleus formation. It would be reassuring to see the same patterns replicated in another cell line or another mechanism of chromothripsis (lagging chromosomes or telomere-deficient chromosome bridges for example).

Referee #3 (Remarks to the Author):

In this manuscript, Lin et al. study how acentric fragments from micronuclei are inherited by daughter cells during mitosis. Using their system to induce micronuclei formation harboring the Y chromosome in a human colon cancer cell line, the authors monitored the partitioning of acentric chromosome fragments, and found that shattered chromosomes from micronuclei cluster together in mitosis, and that CIP2A-TOPBP1 was an essential regulator of this process. CIP2A-TOPBP1 interact with the chromosome fragments upon micronucleus envelope rupture, and this interaction is DNA damage-dependent. Consequences of the mitotic clustering are the inheritance of the shattered chromosome by a single daughter cell, and the suppression of cytoplasmic DNA accumulation in the interphase cytoplasm of that daughter cell. Further, the authors analyzed whole-genome sequencing data from human patients (PCAWG data) and found evidence for high-confidence balanced chromothripsis in ~5% of the tumors. They suggest that these balanced chromothripsis events may result from chromosome fragment clustering, which enables all pieces of a pulverized chromosome to end up in the same daughter cell.

This is a very interesting study that tackles an important and intriguing question: how are acentric chromosome fragments partitioned to daughter cells in mitosis? The reported experiments are well-executed and well-controlled, and key points are supported by multiple types of experiments. The collaboration between the Ly and Cortes-Ciriano groups is powerful as it allows the authors to combine Ly's unique cell biological system with Cortes-Ciriano's unique expertise in analyzing chromothripsis in cancer genomes. In addition, the manuscript is well-written and well-illustrated. I therefore find this manuscript suitable in principle for publication in Nature.

I do have a couple of major comments, however:

(1) A clear limitation of the study is the use of a single cell line (DLD1) and a single chromosome (chromosome Y, which is also the most dispensable chromosome in humans) for many of the experiments. When possible, the authors did use additional HeLa or PC3 cell lines, but the key observation of mitotic clustering is made only in DLD1 cells that were induced to mis-segregate chromosome Y. This raises a concern with regard to the generalizability of the findings, and this concern could be considerably mitigated if the same clustering phenomenon were shown in a different cell line and with a different chromosome. Therefore, I'd strongly encourage the authors to try to demonstrate the clustering phenomenon in at least one additional experimental setting. I am aware that it would be difficult to replicate the DLD1/ChrY system developed by Ly and Cleveland to additional chromosomes. Nonetheless, novel methods for inducing specific chromosome mis-segregation have been recently reported (Klaasen et al. Nature 2022; Truong et al. bioRxiv, <https://doi.org/10.1101/2022.04.19.488790>; Tovini et al. bioRxiv, <https://doi.org/10.1101/2022.04.19.486691>), and it is worth considering whether these methods could be used for monitoring the behavior of micronuclei-captured chromosomes upon micronuclear envelope breakdown.

(2) The authors follow the first cell cycle following micronuclei rupture, and they perform an analysis of cancer genomes in which they identify balanced chromothripsis (which is consistent with the described clustering phenomenon). However, the functional consequences of mitotic clustering are

not studied in the daughter cells that inherit them.

What consequences does mitotic clustering of pulverized chromosomes have for the daughter cells? Would daughter cells of dispersed mitosis (e.g., CIP2A-KO cells) tend to arrest and/or die in the following cell cycle (in comparison to daughter cells of clustered mitoses)? Would there be any difference in DNA damage repair activation? Or in chromosome segregation in the next cell cycles? It would be interesting to continue the live-cell imaging to the next cell cycle(s) and describe how the fate of the daughter cells is affected by the fragment clustering (or in its absence). Such analysis could be a first step towards filling the gap between the cellular phenomenon of mitotic clustering and the genomic observation of balanced chromothripsis (the two culprits of this paper).

Minor comments

(1) Fig. 1g is not referred to in the text (should be referred to along with Fig. 1f).

(2) Can the experiments described in Fig. 2d be quantified? It's impossible to tell otherwise whether the different synchronization methods have any effect.

Author Rebuttals to Initial Comments:

We thank the referees for providing constructive feedback on our manuscript, as well as their positive remarks and enthusiasm. We have considered each point carefully and used it to guide our efforts in improving the study in the revised submission. A detailed, point-by-point response to each comment is included below.

Response to Referee #1:

The manuscript by Lin and colleagues aims to uncover the mechanism that promotes the clustering of micronuclear chromatin fragments that are produced following rupture of micronuclei. They identify the newly described CIP2A-TOPBP1 complex as a mediator of micronuclear chromosome clustering. This mechanism promotes the biased inheritance of the micronuclear genome into a single daughter cell, which preserves copy number and is consistent with observation of balanced chromothripsis, which was first observed in germline chromothripsis. The authors finally provide evidence that balanced chromothripsis is also observed in tumors, and find that the chromosome rearrangements often target oncogenes and tumor suppressors.

Overall, this nicely written manuscript will have a clear impact on our understanding on the formation of micronuclei-driven genome rearrangements in genetic diseases and cancer. Furthermore, the work supports the idea that CIP2A-TOPBP1 acts to tether acentric chromosome fragments, and this cements the role of CIP2A-TOPBP1 in this exciting new biology. Finally, as the CIP2A-TOPBP1 complex was recently shown to be essential in BRCA1/BRCA2-deficient cells, this work provides another reason to consider this complex as a drug target as the loss of micronuclear chromosome clustering is proposed to boost cytosolic DNA sensing, which is a likely anti-tumor mechanism. The data is generally well controlled, uses very elegant approaches and the conclusions are well supported. I am enthusiastic about this work.

I only have a few points that hopefully will improve this already compelling manuscript.

MAIN POINTS

1) The localization of CIP2A and TOPBP1 in ruptured interphase micronuclei raises the possibility that their main function is not related to clustering in mitosis but rather a role in interphase micronuclei (such as DNA repair). Their studies with clonal KOs do not allow the authors to fully exclude this possibility (even if Fig 3e supports their model). Therefore, their conclusions would be strengthened if they could achieve mitotic-specific depletion of CIP2A, TOPBP1 or the CIP2A-TOPBP1 interaction and observe dispersion of micronuclear chromosome fragments in mitotic cells.

We thank the referee for this suggestion. To determine whether CIP2A functions during interphase or mitosis to cluster fragmented chromosomes from micronuclei, we first stably expressed CIP2A fused to a FKBP12^{F36V} degron in CIP2A knockout (KO) DLD-1 cells. We then arrested these cells in mitosis with nocodazole and performed mitotic shake-off to obtain a pure and synchronized population of mitotic cells. In the continued presence of nocodazole, we added the small molecule degrader dTAGv-1 (PMID: 32948771) to induce mitosis-specific depletion of CIP2A-FKBP12^{F36V}, which was confirmed to be efficiently degraded within 4 hours by immunoblotting (**Fig. 2h**). Consistent with a mitotic clustering function, loss of CIP2A during mitosis was sufficient to disperse fragmented micronuclear chromosomes, as determined by IF-FISH (**Fig. 2i-j**). As the referee noted, these data further exclude the repair of these fragments during interphase as an explanation for mitotic clustering.

2) The CIP2A-TOPBP1 complex localizes to mitotic DSBs in an MDC1-dependent manner whereas it localizes to mitotic DNA lesions that arise from under replicated DNA independently of MDC1. The authors should test if the micronuclear CIP2A/TOPBP1 foci are MDC1-dependent, suggesting it recognizes DSBs formed in micronucleus, or whether they form independently of MDC1, perhaps pointing to defective DNA replication as the lesion(s) sensed by CIP2A-TOPBP1.

The referee raises an interesting point in whether MDC1 is involved in CIP2A-TOPBP1-mediated mitotic clustering. We have now tested this using three independent approaches. First, in contrast to a robust fragment dispersion phenotype following depletion of CIP2A or TOPBP1, depletion of MDC1 using three individual siRNAs had a relatively minor effect on dispersing pulverized chromosomes (**Extended Data Fig. 4d**). Second, in the absence of MDC1, both CIP2A and TOPBP1 remained strongly associated with pulverized mitotic chromosomes, as determined by IF-FISH (**Extended Data Fig. 7e-f**). Third, since MDC1 recognizes histone H2AX phosphorylated at serine 139 (γ H2AX), we examined the interphase localization of CIP2A-TOPBP1 in micronuclei in *H2AX*^{-/-} human RPE-1 cells (obtained from PMID: 31729360) and found that γ H2AX is largely dispensable for the recruitment of CIP2A-TOPBP1 to ruptured micronuclei (**Fig. 3b, Extended Data Fig. 6c-d,g**). Altogether, we propose that MDC1 may facilitate – but is not essential for – CIP2A-TOPBP1-mediated mitotic clustering. Furthermore, as the referee highlighted, these data indicate that the underlying DNA lesions sensed by CIP2A-TOPBP1 likely originate from under-replicated DNA, an established hallmark of ruptured micronuclei (PMID: 22258507, 23827674, 26017310). From these new results, we have slightly updated our model in the manuscript to emphasize the recognition of DNA lesions rather than DSBs (**Fig. 3g** and in revised text).

3) The authors show convincingly that micronuclear fragment clustering by CIP2A-TOPBP1 prevents accumulation of cytoplasmic dsDNA. It would be useful to test whether loss of CIP2A, TOPBP1 or the CIP2A-TOPBP1 interaction is also accompanied by stronger or prolonged innate immune response to micronuclei. This may strengthen the case that targeting the CIP2A-TOPBP1 interaction could be therapeutically attractive as an anti-cancer strategy.

To determine whether loss of CIP2A is accompanied by an immune response, we performed RNA sequencing on WT and CIP2A KO HeLa cells (which were chosen over DLD-1 cells that do not intrinsically express the cytosolic DNA sensor cGAS). Gene set enrichment analysis revealed that loss of CIP2A alone was sufficient to activate several pathways that are consistent with an inflammatory response (top panel, **Extended Data Fig. 9f**), which was further heightened in CIP2A KO cells following the induction of micronuclei using CENP-E/Mps1 inhibitors (bottom panel, **Extended Data Fig. 9f**).

We further note that induction of micronuclei in CIP2A KO cells triggered the activation of apoptotic pathways (bottom panel, **Extended Data Fig. 9f**). In agreement with these observations, we tracked the fate of CIP2A-depleted daughter cells generated by the division of micronucleated mother cells and showed that they were more susceptible to cell death during interphase compared to control daughter cells (**Fig. 4k**, also see response to the second comment by referee #3). Thus, in addition to activating an inflammatory response, these data reinforce the notion that CIP2A-deficient cells are selectively vulnerable to the induction of micronuclei.

4) The authors imply that CIP2A-TOPBP1 promotes the formation of balanced chromothripsis over the canonical kind. Testing this prediction experimentally would go a long way towards linking the mechanistic and cancer genomics aspects of the paper.

Referees #1 and #2 raise an important point to functionally link the CIP2A-TOPBP1 complex

to balanced chromothripsis. As previously shown by Zhang *et al.* (Fig. 4b of PMID: 26017310), although most cases of micronucleation result in balanced chromothripsis, it is inferred from single-cell DNA sequencing data that, in some instances, the oscillating DNA copy number patterns characteristic of canonical chromothripsis occurs when pieces of the fragmented chromosome are distributed to both daughter cells (and thus, the fragments that are retained in one daughter cell become lost in the opposite daughter cell and vice versa). To experimentally test whether CIP2A is functionally involved, we performed two sets of new experiments. First, using the FISH assay described in Fig. 1f-g, we measured the distribution of Y chromosome fragments between pairs of newly-divided daughter cells from WT and two CIP2A KO clones. Consistent with the model that mitotic clustering promotes the biased segregation of fragments towards a single daughter cell (as shown in Fig. 1g), we now provide experimental evidence that loss of CIP2A causes the fragments to become stochastically distributed to both daughter cells (Fig. 4a-b).

Second, because mitotic clustering is expected to suppress the loss of genetic material on the derivative chromosome, we assessed the size of rearranged chromosomes from WT and CIP2A KO cells. To do so, we measured the frequency of Y chromosome rearrangements using a previously characterized FISH approach leveraging two paint probes that recognizes the first and second half of the Y chromosome, which allows rearrangements to be visibly discernable by cytogenetics (Fig 4c). Analysis of 2,934 metaphase spreads revealed that cells lacking CIP2A developed significantly fewer complex rearrangements of the Y chromosome (Fig 4d-e), consistent with inefficiencies in repairing fragments that are dispersed throughout daughter cell nuclei (in contrast to clusters of fragments that are localized to a confined region in the nucleus). Since these complex rearrangements were rare, we then leveraged a neomycin-resistance marker encoded on the Y chromosome to isolate a number of Y chromosomes harboring complex rearrangements from WT and CIP2A KO cells for further analysis (Fig 4e). We measured the size of rearranged Y chromosomes by quantifying the number of pixels occupied by DAPI staining, and to account for variation in chromosome size between different metaphase spreads, we normalized these data relative to the corresponding X chromosome per cell. Consistent with the model that mitotic clustering restricts DNA copy number losses, Y chromosomes with complex rearrangements were often smaller in size in CIP2A KO cells compared to complex rearrangements derived from WT controls (Fig 4f). These new experiments provide a functional link between CIP2A and copy number outcome on the pulverized chromosomes.

Response to Referee #2:

The authors present an analysis of the role of the CIP2A-TOPB1 complex in maintaining chromosomal fragments in close proximity to one another. One prediction of the model is that there should be examples of chromothripsis in which all chromosomal fragments are retained in the eventual reconstructed chromosome, and indeed the authors find examples of so-called 'balanced chromothripsis' in which there are no copy number changes despite the presence of multiple SVs suggestive of chromothripsis.

I enjoyed reading this manuscript, and there is much to like in the dissection of the role of CIP2A-TOPB1 - I think this story is of interest to the general readership of Nature. I have two major comments that would strengthen the story –

1. The most important limitation of the story as currently presented, in my opinion, is the lack of a link between the human data and the CIP2A-TOPB1 complex story - that is, the existence of balanced chromothripsis in human cancers does not substantiate the relevance of that complex to chromothripsis *in vivo*. More convincing evidence could come in the form of either (1) an analysis that showed that unbalanced chromothripsis was more common in cancers that had LOH / mutation of the complex or (2) an *in vitro* study of the DLD-1 cell line

showing that the chromothripsis that emerged after CIP2A/TOPBP1 knockdown was more likely to be unbalanced than when the complex is intact.

Referees #1 and #2 raise an important point to functionally link the CIP2A-TOPBP1 complex to balanced chromothripsis. We have now performed a series of new experiments to address this. We would like to kindly refer the reviewer to our response to the fourth comment by referee #1.

Additionally, as the referee suggested, we interrogated the PCAWG datasets for LOH/mutations in CIP2A and TOPBP1, and we found that both of these genes were infrequently mutated across diverse cancer types. This analysis is further underpowered when considering that chromothripsis is only detected within 25-30% of cancer genomes (PMID: 32025003). The highest rate of mutations in these genes were actually found in tumors where hypermutation is frequent (e.g., MSI), indicating that many of these mutations are likely passenger events arising throughout tumorigenesis.

2. The in vitro analysis is restricted to a single cell line with one mechanism of micronucleus formation. It would be reassuring to see the same patterns replicated in another cell line or another mechanism of chromothripsis (lagging chromosomes or telomere-deficient chromosome bridges for example).

We completely agree with this point raised by referees #2 and #3. As suggested, we have now extended the key findings of our study into additional cell models, methods of inducing chromosome mis-segregation, and chromosomes beyond the Y chromosome:

First, we transiently arrested hTERT-immortalized, non-transformed human retinal pigment epithelial (RPE-1) cells in mitosis with nocodazole followed by washout and release into interphase. Prior work has shown that this procedure causes a high frequency of non-random chromosome 1 segregation errors in the form of lagging chromosomes (PMID: 29898405). We validated this approach and confirmed that WT and CIP2A KO RPE-1 cells generated chromosome 1-specific micronuclei at identical frequencies (**Extended Data Fig. 5b-d**). Second, we introduced a DNA DSB onto chromosome 3p in human renal proximal tubule epithelial cells (RPTECs) by transfection with Cas9 RNP in the presence of a DNA-PK inhibitor to suppress DSB repair. When left unrepaired into mitosis, the acentric chromosome 3p arm mis-segregates into micronuclei. Prior work has established that micronuclei generated using CRISPR/Cas9 can indeed trigger chromothripsis (PMID: 33846636). Using both of these cell lines and distinct strategies to generate micronuclei, we now show that inactivation of CIP2A is sufficient to disrupt mitotic clustering and produce visible chromosome 1 and chromosome 3p fragments in RPE-1 cells (**Extended Data Fig. 5f-g**) and RPTECs (**Extended Data Fig. 5j-k**), respectively, that were otherwise nearly undetectable under control conditions. Additionally, we have extended our analyses of interphase CIP2A-TOPBP1 recruitment to micronuclei to include RPE-1 and RPTECs (**Fig. 3b and Extended Data Figure 6**) in the revised manuscript.

Response to Referee #3:

In this manuscript, Lin et al. study how acentric fragments from micronuclei are inherited by daughter cells during mitosis. Using their system to induce micronuclei formation harboring the Y chromosome in a human colon cancer cell line, the authors monitored the partitioning of acentric chromosome fragments, and found that shattered chromosomes from micronuclei cluster together in mitosis, and that CIP2A-TOPBP1 was an essential regulator of this process. CIP2A-TOPBP1 interact with the chromosome fragments upon micronucleus envelope rupture, and this interaction is DNA damage-dependent. Consequences of the mitotic clustering are the inheritance of the shattered chromosome by a single daughter cell, and the suppression of cytoplasmic DNA accumulation in the interphase cytoplasm of that

daughter cell. Further, the authors analyzed whole-genome sequencing data from human patients (PCAWG data) and found evidence for high-confidence balanced chromothripsis in ~5% of the tumors. They suggest that these balanced chromothripsis events may result from chromosome fragment clustering, which enables all pieces of a pulverized chromosome to end up in the same daughter cell.

This is a very interesting study that tackles an important and intriguing question: how are acentric chromosome fragments partitioned to daughter cells in mitosis? The reported experiments are well-executed and well-controlled, and key points are supported by multiple types of experiments. The collaboration between the Ly and Cortes-Ciriano groups is powerful as it allows the authors to combine Ly's unique cell biological system with Cortes-Ciriano's unique expertise in analyzing chromothripsis in cancer genomes. In addition, the manuscript is well-written and well-illustrated. I therefore find this manuscript suitable in principle for publication in Nature.

I do have a couple of major comments, however:

(1) A clear limitation of the study is the use of a single cell line (DLD1) and a single chromosome (chromosome Y, which is also the most dispensable chromosome in humans) for many of the experiments. When possible, the authors did use additional HeLa or PC3 cell lines, but the key observation of mitotic clustering is made only in DLD1 cells that were induced to mis-segregate chromosome Y. This raises a concern with regard to the generalizability of the findings, and this concern could be considerably mitigated if the same clustering phenomenon were shown in a different cell line and with a different chromosome. Therefore, I'd strongly encourage the authors to try to demonstrate the clustering phenomenon in at least one additional experimental setting.

I am aware that it would be difficult to replicate the DLD1/ChrY system developed by Ly and Cleveland to additional chromosomes. Nonetheless, novel methods for inducing specific chromosome mis-segregation have been recently reported (Klaasen et al. Nature 2022; Truong et al. bioRxiv, <https://doi.org/10.1101/2022.04.19.488790>; Tovini et al. bioRxiv, <https://doi.org/10.1101/2022.04.19.486691>), and it is worth considering whether these methods could be used for monitoring the behavior of micronuclei-captured chromosomes upon micronuclear envelope breakdown.

We completely agree with this point raised by referees #2 and #3. As suggested, we have now extended the key findings of our study into additional cell models, methods of inducing chromosome mis-segregation, and chromosomes beyond the Y chromosome. We would like to kindly refer the reviewer to our response to the second comment by referee #2.

(2) The authors follow the first cell cycle following micronuclei rupture, and they perform an analysis of cancer genomes in which they identify balanced chromothripsis (which is consistent with the described clustering phenomenon). However, the functional consequences of mitotic clustering are not studied in the daughter cells that inherit them. What consequences does mitotic clustering of pulverized chromosomes have for the daughter cells? Would daughter cells of dispersed mitosis (e.g., CIP2A-KO cells) tend to arrest and/or die in the following cell cycle (in comparison to daughter cells of clustered mitoses)? Would there be any difference in DNA damage repair activation? Or in chromosome segregation in the next cell cycles? It would be interesting to continue the live-cell imaging to the next cell cycle(s) and describe how the fate of the daughter cells is affected by the fragment clustering (or in its absence). Such analysis could be a first step towards filling the gap between the cellular phenomenon of mitotic clustering and the genomic observation of balanced chromothripsis (the two culprits of this paper).

We appreciate this suggestion from the referee. To determine the consequences of mitotic clustering of pulverized chromosomes for the daughter cells, we used extended live-cell

imaging to monitor the fate of newly-formed daughter cells generated by the division of micronucleated mother cells. Daughter cells lacking CIP2A were noticeably less fit and died four times more frequently during the subsequent interphase compared to their control counterparts (**Fig. 4k**). This was further confirmed by single-cell clonogenic growth assays in which CIP2A-deficient cells formed significantly fewer colonies than WT cells upon the induction of micronuclei (**Extended Data Fig. 9g**). These new data demonstrate that CIP2A loss renders vulnerability to the induction of micronuclei. Additionally, to determine whether there would be any differences in DNA damage repair activation, we immunostained WT and CIP2A KO cells for 53BP1, an established marker of DNA repair, in combination with Y chromosome FISH. Indeed, 53BP1 localized to both clustered and dispersed chromosome fragments in the nucleus during interphase (**Extended Data Fig. 9a-b**), suggesting that dispersed fragments are capable of engaging the DNA damage response similarly to clustered fragments.

Minor comments

(1) Fig. 1g is not referred to in the text (should be referred to along with Fig. 1f).

We apologize for this oversight, which has now been corrected in the revised text.

(2) Can the experiments described in Fig. 2d be quantified? It's impossible to tell otherwise whether the different synchronization methods have any effect.

We have now included quantification for these experiments. Dispersion frequencies for each condition are shown on the corresponding images along with the sample size (**Fig. 2d**).

Reviewer Reports on the First Revision:

Referees' comments:

Referee #1 (Remarks to the Author):

The authors have done a great job addressing my previous comments. I am enthusiastic about this work.

Referee #2 (Remarks to the Author):

The authors have comprehensively addressed my comments and I remain enthusiastic about the manuscript.

Referee #3 (Remarks to the Author):

The authors have fully addressed my comments, as well as those of the other referees. I have no further requests, and I support the acceptance of the manuscript to Nature. I'd like to congratulate the authors for their great work.